

# Impacts of future deforestation and climate change on the hydrology of the Amazon basin: a multi-model analysis with a new set of land-cover change scenarios

Matthieu Guimberteau[1], Philippe Ciais[1], Agnès Ducharne[2], Juan Pablo Boisier[3], Ana Paula Dutra Aguiar[4], Hester Biemans[5], Hannes De Deurwaerder[6], David Galbraith[7], Bart Kruijt[5], Fanny Langerwisch[8], German Poveda[9], Anja Rammig[8,10], Daniel Andres Rodriguez[11], Graciela Tejada[4], Kirsten Thonicke[8], Celso Von Randow[4], Rita C. S. Von Randow[4], Ke Zhang[12,13,14], Hans Verbeeck[6]

[1]Laboratoire des Sciences du Climat et de l'Environnement, LSCE/IPSL, CEA-CNRS-UVSQ, Université Paris-Saclay, F-91191 Gif-sur-Yvette, France

[2]Sorbonne Universités, UPMC, CNRS, EPHE, UMR 7619 METIS, 75252 Paris, France

[3]Department of Geophysics, Universidad de Chile, and Center for Climate and Resilience Research (CR2), Santiago, Chile

[4]Centro de Ciência do Sistema Terrestre (CCST), Instituto Nacional de Pesquisas Espaciais (INPE), Av dos Astronautas 1758, 12227-010, São José dos Campos, Brazil

[5]Climate Change and Adaptive Land and Water Management Group, Alterra, Wageningen University and Research, P.O. Box 47, 6700 AA Wageningen, Netherlands

[6]CAVElab – Computational and Applied Vegetation Ecology, Department of Applied Ecology and Environmental Biology, Faculty of Bioscience Engineering, Ghent University, Coupure Links 653, 9000 Ghent, Belgium

[7]School of Geography, University of Leeds, Leeds, United-Kingdom

[8]Earth System Analysis, Potsdam Institute for Climate Impact Research (PIK), P.O. Box 60 12 03, Telegraphenberg A62,14412 Potsdam, Germany

[9]School of Geosciences and Environment, Universidad Nacional de Colombia, Medellín, Colombia

[10]TUM School of Life Sciences Weihenstephan, Land Surface-Atmosphere Interactions, Technical University of Munich, Freising, Germany

[11]Centro de Ciência do Sistema Terrestre (CCST), Instituto Nacional de Pesquisas Espaciais (INPE), Rodovia Presidente Dutra km 39, CP 01,CEP: 12630-000, Cachoeira Paulista, São Paulo, Brazil

[12]Department of Organismic and Evolutionary Biology, Harvard University, Cambridge, MA, USA

[13]Cooperative Institute for Mesoscale Meteorological Studies, University of Oklahoma, Norman, OK, USA

[14]College of Hydrology and Water Resources, Hohai University, Nanjing, Jiangsu Province, China

*Correspondence to:* M. Guimberteau (matthieu.guimberteau@lsce.ipsl.fr)

**Abstract.** Neglecting any atmospheric feedback to precipitation, deforestation in Amazon, i.e., replacement of trees by shallow rooted short vegetation, is expected to decrease evapotranspiration (ET). Under energy-limited conditions, this process should lead to higher soil moisture and a consequent increase in river discharge. The magnitude and sign of the response of ET to deforestation depends both on land-cover change (LCC), and on climate and $CO_2$ concentration changes in the future. Using





three regional LCC scenarios recently established for the Brazilian and Bolivian Amazon, we investigate the combined impacts of deforestation and climate change on the surface hydrology of the Amazon basin for this century at sub-basin scale. For each LCC scenario, three land surface models (LSMs), LPJmL-DGVM, INLAND-DGVM and ORCHIDEE, are forced by bias-corrected climate simulated by three General Circulation Models (GCMs) for different scenarios of the IPCC 4th Assessment

Report (AR4). The GCM results indicate that by 2100, without deforestation, the temperature will have increased by a mean of 3.3°C (a range of 1.7 to 4.5°C) over the Amazon basin, intensifing the regional water cycle, whereby precipitation, ET and runoff increase by 8.5, 5.0 and 14%, respectively. However, under this same scenario in south-east Amazonia, precipitation decreases by 10% at the end of the dry season and the three LSMs estimate a 6% decrease of ET, which does not compensate for lower precipitation. Runoff in southeastern Amazonia decreases by 22%, reducing minimum river discharge from the Rio

Tapajós catchment by 31% in 2100. The low LCC scenario projects a 7% decline in the area of Amazonian forest by 2100, as compared to 2009; for the high LCC scenario the projection is a 34% decline. In the extreme deforestation scenario, forest loss partly offsets (-2.5%) the positive effect of climate change on increasing ET and slightly amplifies (+3.0%) the increase of runoff. Effects of deforestation are more pronounced in the southern part of the Amazon basin, in particular in the Rio Madeira catchment where up to 56% of the 2009 forest area is lost. The effect of deforestation on water budgets is more severe at the

end of the dry season in the Tapajós and the Xingu catchments where the decrease of ET due to climate change is amplified by forest area loss. Deforestation enhances runoff during this period (+35%) offsetting the negative effect of climate change (-22%), and balances the decrease of low flows in the Rio Tapajós.

**Keywords:** Amazon basin, hydrology, land-cover change, land surface models

# 1 Introduction

The Amazon basin provides a range of ecosystem services. The rivers are used for navigation and hydropower; the forest is an important global sink and store of carbon, and a store of biodiversity; evaporation provides a water vapour source for rainfall downwind. When analysing changes to this ecosystem, it is important to take an integrated approach because each of these services may be affected by, or may affect, the others. Currently, two major changes are taking place simultaneously in Amazonia: deforestation and climate change. From the early 1970s, southern Amazonia has experienced widespread deforestation with

forest being cleared to create new pasture and cropland (Fearnside, 2005). About 7.3% of the Amazon basin was deforested between 1976 and 2003 (Callède et al., 2008) and a further 2.6% between 2000 and 2010 (Song et al., 2015). At the same time, the background level of $CO_2$ has been rising and the climate has been changing in response. These changes are expected to continue, to some degree, for the rest of this century.

Here, we focus on future changes to the river hydrology of the Amazon basin. For different deforestation scenarios, we

model the changes in river flow from drainage-and-runoff estimated by land surface models (LSMs) driven by forcing data derived from general circulation model (GCM) output. Because of the long transit times of water moving from soil to the mouth of the Amazon, to simulate discharge requires LSMs to be coupled to a river routing scheme (Biemans et al., 2009; Guimberteau et al., 2012; Langerwisch et al., 2013).





The climate of the Amazon basin is notoriously difficult to model and there is a wide between-GCM variation in the estimated precipitation and its changes (Boisier et al., 2015). This introduces a first level of uncertainty.

Equally, several LSMs exist and, to a greater or lesser extent, they all incorporate existing process knowledge into their parameterizations (Gash et al., 2004; Keller et al., 2009). However, because of their different structures and the values of the parameters used, LSMs also simulate a range of changes in the water and energy balances, produced by changes in vegetation and its structure, phenology and physiology. This introduces a second level of uncertainty.

A third level of uncertainty stems from the development scenarios used. Observed historical deforestation rates in Amazonia are substantially higher than future rates projected in the SSPs (Shared Socio-economic Pathways) of global scenarios (Representative Concentration Pathways, RCPs) used for the last CMIP (Coupled Model Intercomparison Project) assessment. This disparity questions the realism of the projected rates at regional scales (Soares-Filho et al., 2006; Kay et al., 2013). On the other hand, deforestation can also be reduced by conservation policies, and none of the previous LCC (land-cover change) modelling exercises for Amazonia were able to plausibly anticipate the 84% decrease in deforestation rates in Brazil during the last decade (INPE, 2012; Dalla-Nora et al., 2014). The SSPs probably fail to capture the trajectory of regional LCC because they do not integrate regional land management policies, existing or future road building, or the establishment of conservation areas (Dalla-Nora et al., 2014).

Here, we introduce more realistic LCC scenarios combining these in a three dimensional matrix of model simulations to combine the effect of uncertainty in forcing data, LSMs and development scenarios. This matrix of combinations allows us to estimate the magnitude of likely future hydrological changes due to deforestation and climate change and their uncertainty.

## 2 Materials and methods

### 2.1 Simulation design and models

The time frame studied includes an historical period representing current climate conditions (1970-2008) and 21st century projections (2008-2100). Although the domain used in the simulations described below includes the whole Amazon basin (Fig. 1a and Table S1), the analysis focuses on the sub-basins sensitive to deforestation (Fig. 1b). We selected the southern sub-basins, which are subjected to a distinct dry season today, and are both sensitive to future precipitation changes (Guimberteau et al., 2013; Boisier et al., 2015) and vulnerable to future deforestation (Coe et al., 2009; Costa and Pires, 2010). These catchments are the Rio Madeira (MAD) and its upstream tributary, the Mamoré (MAM), and the two large south-eastern catchments of the Tapajós (TAP) and Xingu (XIN) (Fig. 1a). We also chose three western catchments, the Purus (PUR), Juruá (JUR) and Upper Solimões (UPSO), and the northern Rio Branco catchment (BRA). These catchments have also experienced deforestation (Nóbrega, 2012; Lima et al., 2014; Barni et al., 2015). The river discharge of the Amazon basin is taken from gauging station at Óbidos. Although this station is the closest to the mouth of the Amazon, it is upstream from the confluence of the Tapajós and Xingu with the main stem of the Amazon (Fig. 1a). The Óbidos data therefore does not contain the contribution of these rivers.



We used three LSMs, namely LPJmL-DGVM and INLAND-DGVM, which simulate daily water budgets interactively with changes in vegetation physiology, and ORCHIDEE, which operates with a 30-min time-step (see Table 1 and description in supplementary material).

First, we performed an historical simulation (1970-2008, HIST, Table 2) where we forced the LSMs with pre-industrial land cover and the Princeton global climate (Sheffield et al., 2006) at a $1° \times 1°$ spatial resolution and 3-hourly temporal resolution (Fig. 2). This forcing is based on the National Center for Environmental Prediction-National Center for Atmospheric Research (NCEP-NCAR) 6-hourly reanalysis data sets (Kistler et al., 2001) with precipitation, air temperature and radiation biases corrected by hybridization with global monthly gridded observations. The corrected precipitation was disaggregated in space by a statistical downscaling at $1°$ resolution using relationships developed by the Global Precipitation Climatology Project (GPCP, Huffman et al., 2001), and in time from daily to 3-hourly using the Tropical Rainfall Measuring Mission (TRMM; Huffman et al., 2007) satellite data. A 300-year spin-up was performed by each LSM to ensure equilibrium of carbon and water pools by recycling the Princeton forcing over the period 1970-2008 with constant pre-industrial atmospheric $CO_2$ concentration representative of the year 1970 (278 ppm). Starting from the end of the spin-up state, all the LSMs did the simulation HIST forced with increasing $CO_2$ (from 278 ppm to 385 ppm) derived from atmospheric observations and with observed climate change. Neither the spin-up nor the HIST runs account for LCC. Each LSM used its own definition of natural land cover and soil parameters (Table 1).

Using HIST as initial conditions, multiple future simulations with each LSM were run from 2009 to 2100 (Table 2, Fig. 2) with increasing $CO_2$ from the SRES A2 scenario (388 ppm to 856 ppm). For the atmospheric forcing, we used three GCMs from CMIP3 (Table 3), bias-corrected and regridded to $1°$ as detailed in Zhang et al. (2015) and Moghim et al. (2016) (Fig. 2). First, to define the hydrological response to climate change only, we performed a future simulation with land cover set constant at the value for year 2009 for each LSM (NODEF, Table 2). Then, in order to separate the impacts of future deforestation, we prescribed to each LSM three annual LCC spatial projections (Fig. 2), resulting from the scenarios of Aguiar et al. (2016) and generated in the scope of the AMAZALERT project (Raising the alert about critical feedbacks between climate and long-term land use change in the Amazon, http://www.eu-amazalert.org/home).

## 2.2 Deforestation scenarios

These three regional LCC scenarios were generated using a qualitative/quantitative participatory approach (Aguiar et al., 2016). Future maps of forest area were simulated using the LuccME (Land use and cover change Modeling Environment, http://www.terrame.org/luccme) model framework, generating annual forest cover maps until 2100 on a grid of 25 $km^2$ for the Brazilian Amazon (Aguiar et al., 2016) and the Bolivian Amazon (Tejada et al., 2015). To generate basin-wide LCC projections, the annual spatially explicit results for Brazil and Bolivia were combined with the existing "business-as-usual" projection to the other countries, based on historical deforestation trends spatialized using the CLUE (Conversion of Land Use and its Effects) model (Verburg et al., 2002), as part of the EU-funded ROBIN (Role Of Biodiversity In climate change mitigatioN) project (Eupen et al., 2014).



Scenario A is a "Sustainability" scenario in terms of socioeconomic, institutional and environmental dimensions, with no deforestation and massive forest restoration. Scenario B stays in the "Middle of the road", maintaining some of the positive trends of the last decade (in the case of Bolivia the deforestation rate increased in this period), but not reaching the full potential from an integrated socioeconomic, institutional and environmental perspective. Finally, Scenario C is a pessimistic scenario,

named "Fragmentation", consisting of a weakening of the conservation efforts of recent years, including the depletion of natural resources and the return of high deforestation rates. Although the participatory process was only carried out in Brazil, similar assumptions were adopted for Bolivia (Tejada et al., 2015). Then, based on a coherent set of premises about the land-use dynamics extracted from the storylines, the scenarios were quantified using the LuccME modelling framework in order to build models for the Brazilian (LuccME/BRAmazon) and Bolivian (LuccME/Bolivia) Amazon.

LuccME (Aguiar et al., 2012) is a generic framework to build land-use demand-potential-allocation models. In general terms, land change decisions are controlled by an allocation mechanism which uses the suitability of each cell for a given land change transition (potential of change) to distribute a given amount (demand) of change in space. LuccME allows the construction of LUCC models combining existing Demand, Potential and Allocation components according to the needs of a given application and scale of analysis. The modelling components adopted to build the LuccME/BRAmazon and LuccME/Bolivia models

work, in general lines, as follows. Cells with positive change potential will receive a percentage of the annual deforestation rate expected to be allocated to the whole area. The amount of change (i.e., new deforestation) in each cell will be proportional to the cell potential, which is recomputed, every year, considering not only the temporal changes in the spatial drivers (for instance, creation/extinction of protected areas, building/paving roads), based on the scenario premises, but also the distance to previously opened areas.

For the Brazilian Amazon, annual spatially-explicit deforestation maps from 2002-2013, provided by the PRODES (Program for the Estimation of Deforestation in the Brazilian Amazon) system (INPE, 2016), were used to calibrate and validate the parameters of the deforestation model (LuccME/Bolivia). The scenario projections generated for the AMAZALERT project cover the period 2014 to 2100. The same premises described by Aguiar et al. (2016) were used to extend the projections from 2050 to 2100. The modelling process for the Bolivian Amazon (LuccME/Bolivia) was similar. The observed deforestation data

were drawn from the NKMMNH (Noel Kempff Mercado Museum of Natural History, Killeen et al., 2012) from 2001–2008. In this case, the scenario projections run from 2009-2100, adapting the premises described by Tejada et al. (2015) from 2050-2100.

In this paper, we explore the effects of Scenario A and Scenario C contrasting storylines. Scenario A storyline quantification produced low forest loss (LODEF), whilst Scenario C was quantified into a high (HIDEF) and extreme (EXDEF) forest area

loss for the Brazilian Amazon (Table 4). Fig. S2 illustrates the basin-wide maps, combining the LuccME/BRAmazon, the LuccME/Bolivia and the other countries' spatial projections (Eupen et al., 2014). As Fig. S2 illustrates, the results were combined into a 25 x 25 km$^2$ grid cell for the whole basin, containing annual information (from 2005 to 2100) about the percentage of the cell area that was deforested up to that year.

These three LCC scenarios (LODEF, HIDEF and EXDEF) were translated into model parameters and prescribed to each

LSM (Table 2, Fig. 2) from 2009 to 2100.





## 2.3 Model results analysis

We selected two 20-year periods, 2040–2059 and 2080–2099, for LSM output analysis. The impact of future climate change alone was estimated for each LSM by the difference between the results of NODEF and HIST in precipitation, ET, runoff and river discharge (Table 2). The impact of future LCC was estimated by taking the difference, for each LSM and GCM-forcing,

between the results of future simulations with LCC (LODEF, HIDEF and EXDEF) and without LCC (NODEF).

The spread in the ensemble mean variation (LSMs and GCM-forcings) was measured by the interquartile range (IQR). The consistency of the variations in precipitation, ET and runoff were estimated by the signs of the first ($Q_1$) and the last ($Q_3$) quartiles. A decrease (increase) was considered to be consistent if $Q_3 < 0$ ($Q_1 > 0$).

## 3 Results

### 3.1 Future scenarios

### 3.1.1 Climate

By the end of the 21st century, GCM-mean annual temperature increases by 3.3°C in Amazonia. The forcing from the GCM chosen in this study spans the range of climate predictions for Amazonia (Malhi et al., 2009; Zhang et al., 2015). The UKMO-HadCM3 GCM is the driest and warmest model (+4.5°C), predicting Amazon rainfall reductions twice as large as any other

CMIP3 GCM (Covey et al., 2003; Cox et al., 2004); PCM simulates a slightly warmer (+1.7°C) but wetter future climate compared to the current climate, and CCSM3 falls in-between (+3.6°C) (Zhang et al., 2015). The strongest warming of 6.1°C by 2100 is found in eastern Amazonia with the UKMO-HadCM3 GCM (not shown). Because of the differences in precipitation changes projected by the three GCMs (between -4.5 and +16.2%) by the end of the century, the average increase of precipitation by 8.5% (190 mm yr$^{-1}$) across the three GCMs should be considered as very uncertain (Fig. 3a). Precipitation changes are also

spatially contrasted. Western and northern Amazonia tend to become wetter in all GCM models, with annual precipitation increases going from 6.5 to 11% (Figs 3a and 4a). In the Upper Solimões and the Branco catchments, the three GCM-forcings give an increase of precipitation. Southern Amazonia also becomes wetter, in particular the Madeira catchment where the three GCM-forcings give a 5% increase in precipitation (Fig. 3a) but with spatial differences. In the Madre de Dios region (see Fig. 1 for location), at least two out of the three GCM-forcings give a decrease in precipitation (Fig. 4a). In south-eastern Amazonia,

there is no change in GCM-mean precipitation over the Tapajós and Xingu catchments but GCM-forcings disagree on the sign and magnitude of the change (Fig. 3a). At least two out of the three GCM-forcings give a decrease of precipitation in the western part of the Tapajós catchment (Fig. 4a).

We focus on the period corresponding to the end of the dry season, from August to October (ASO). Lower precipitation during this period could have critical effects on the vegetation and hydrology (Malhi et al., 2008). In ASO, the spatial patterns

of precipitation changes are similar to annual mean changes, but with a larger area of decreased precipitation (Figs. 4a and 4b). As noticed by Guimberteau et al. (2013), south-eastern Amazonia becomes drier in the middle and at the end of the




century, with an average ASO precipitation decrease ranging from 10 to 14% in the Tapajós and Xingu catchments (Fig. 3b). At least two out of the three GCM-forcings give a consistent ASO precipitation decrease in most of the grid cells of these two catchments, particularly in the Xingu (Fig. 4b). In other southern regions, GCM-forcings predict wetter ASO conditions with a larger precipitation increase simulated by UKMO-HadCM3 in the southernmost part of the Madeira catchment compared

with the two other GCM-forcings (Fig. 4b). In the western catchments, UPSO, PUR and JUR, and the northern BRA, all the GCM-forcings predict a consistent precipitation increase during ASO of between 16 and 30% (3b), except in the northernmost part of the Upper Solimões catchment (Fig. 4b).

### 3.1.2 Land-cover change

The total area of Amazonian forest prescribed in 2009 is 5.27 Mkm$^2$, i.e., 89% of the total area of the whole basin (Fig.

5). The LODEF scenario projects a 7% decrease in forest area over the Amazon basin by 2099 relative to 2009 (Fig. 6). By comparison, the SSP land-use scenario with the RCP8.5 emission scenario, which broadly corresponds to the SRES A2 storyline of the GCM climate forcing used in this study, gives a forest area loss of 4.6% (Fig. 5), relative to a forest area of 5.03 Mkm$^2$ in 2009. By contrast, in both HIDEF and EXDEF, forest area strongly declines during the next century. By 2100, according to the EXDEF scenario, the area of Amazonian forest is reduced to a value of 3.45 Mkm$^2$ (34%) (Figs 5 and

6), i.e., about half of the Amazon basin. LCC scenarios show a high heterogeneity at the resolution of 0.25°, reflecting how fragmentation is simulated in the LuccME model (Fig. 7). The southern catchments experience the highest deforestation rates during the 21$^{st}$ century in all scenarios. The Madeira catchment loses some 14% of its forest area by 2100 in LODEF, and more than 50% in HIDEF and EXDEF (Fig. 6). In the Mamoré southern sub-catchment of the Madeira, forest area loss reaches 60% in HIDEF and EXDEF (Fig. 6) with the deforested area reaching 100% in the upstream part of MAM (Fig. 7f). In the southern

Tapajós and Xingu catchments, LODEF and EXDEF give contrasting estimates of forest area: in LODEF, forest area changes by only 1% in 2099 whereas in EXDEF it decreases by approximately 50% (Fig. 6). The western and northern catchments are projected to lose between 2 and 40% of their forest area, depending on the LCC scenario.

### 3.2 Effects of climate change on evapotranspiration and runoff

### 3.2.1 Annual mean changes in ET and runoff

The 8.5% average increase of GCM-estimated annual precipitation (190 mm yr$^{-1}$) by the end of the century results in a 5% increase in ET (54 mm yr$^{-1}$) and a 14% increase in runoff (136 mm yr$^{-1}$) over the entire Amazon basin (Figs 3a, 3c and 3e, respectively). The ensemble spread in annual ET variation is lower (IQR = 110 mm yr$^{-1}$, Fig. 3c) than the change in runoff (IQR = 420 mm yr$^{-1}$), which is very uncertain (Fig. 3e). Western parts of the basin become wetter, and ensemble-mean ET and runoff consistently increase, by up to 6.5 and 16%, respectively, with a higher spread for runoff variation (IQR > 250 mm yr$^{-1}$).

The largest increase of ensemble-mean runoff (19%) in the Amazon basin, occurs in the northern catchments where the increase in ET is the smallest (<2%). This increase is associated with a large spread between the multiple simulations (IQR close to 600 mm yr$^{-1}$). In southern parts of the basin, the 5% increase of the ensemble-mean ET is uncertain when considering the




Mamoré and the south-eastern sub-catchments (Fig. 3c). Yet, in the foothills of the Andes, the northern Madeira sub-catchment and along the Amazon River (Fig. 8a), the increase of ET is consistent across the multiple simulations. In the northwestern part of the Madeira sub-catchment (MAM, Beni and Madre de Dios rivers) with an annual precipitation decrease (Fig. 4a), six out of nine simulations show decrease in ET (Fig. 8a). In the Mamoré sub-catchment, mean-ensemble runoff consistently

increases by 23% whereas it decreases by 6% in the south-eastern sub-catchments. The runoff changes spread widely across all the simulations in these catchments (IQR $\approx 430\,\mathrm{mm\,yr^{-1}}$, Fig. 3e).

### 3.2.2 South-eastern catchments: ASO changes in ET and runoff

During ASO, the end of the dry season in the south-eastern catchments, reduced precipitation causes a consistent decrease in ET, e.g., in the Xingu catchment by up to 8% ($10\,\mathrm{mm\,month^{-1}}$, Fig. 3d), where at least six out of nine simulations give a

consistent ET reduction in many southern grid cells (Fig. 8b). The spread between all the projections simulating ET decrease is lower for the Tapajós than for the Xingu (IQR is 21 and $27\,\mathrm{mm\,month^{-1}}$, respectively). Mean-ensemble runoff, already low during ASO in this region, decreases consistently by about 25% and the ensemble spread is low (IQR $< 10\,\mathrm{mm\,month^{-1}}$).

### 3.3 Effects of deforestation (with background climate change)

### 3.3.1 Annual mean changes in ET and runoff

Deforestation and climate change lead to a consistent decrease in annual ET in the Amazon basin by the end of the century, of up to 2.5% ($30\,\mathrm{mm\,yr^{-1}}$) with the EXDEF scenario (Fig. 9a). The resulting consistent increase of runoff is less than 3% (Fig. 10a) and both spreads of ET and runoff over the entire basin are small between the multiple forcings and LSMs used (IQR = $30\,\mathrm{mm\,yr^{-1}}$ for runoff). With the EXDEF scenario, the loss of forest area leads to a continuous ET reduction throughout the 21$^{\mathrm{st}}$ century but of different magnitude depending on the simulation type (Fig. 11a). By the end of the 21st century,

ORCHIDEE simulates a $58\,\mathrm{mm\,yr^{-1}}$ ET reduction while LPJmL-DGVM gives a decrease of $12\,\mathrm{mm\,yr^{-1}}$ (multi GCM-forcing mean). In addition, for a given LSM, the decrease of ET differs according to the GCM-forcing used, notably in the case of ORCHIDEE, that simulates a decrease twice as large with UKMO-HadCM3 than with PCM. In EXDEF, ET is more strongly affected in the southern and eastern regions where forest area loss is important (compare Fig. 7f and Fig. 8c). A decrease in ET by up to $\sim 150\,\mathrm{mm\,yr^{-1}}$ is obtained in these regions, where at least six out of nine simulations show a consistent ET decrease in

the EXDEF scenario. In south-eastern catchments, the strong reduction of forest area in EXDEF leads to a consistent reduction of annual ET by 7% ($\sim 80\,\mathrm{mm\,yr^{-1}}$, Fig. 9a) and to a consistent increase in runoff by 10% (Fig. 10a). In these two catchments, the spread within the ensemble is higher than in other regions (IQR = $80\,\mathrm{mm\,yr^{-1}}$ for runoff increase in TAP). In the western and northern catchments, deforestation and climate change induce a maximum consistent ET (runoff) reduction (increase) of less than 5% (6%) in each sub-catchment.



### 3.3.2 South-eastern catchments: ASO changes in ET and runoff

During ASO in the south-eastern catchments, ensemble-mean ET consistently decreases by up to 12% and runoff increases by 35% in the EXDEF scenario (Figs 9b and 10b). ORCHIDEE and INLAND-DGVM simulate the highest ET decreases in August over the Tapajós (by 23 and 12 mm month$^{-1}$, respectively) while ET decreases most in October in LPJmL-DGVM (-

6.0 mm month$^{-1}$) (Fig. 12c). Deforestation reduces both the evaporation of intercepted rainfall and transpiration, by up to 45% during the wet season in ORCHIDEE (Fig. S2b), and increases soil evaporation by the same order of magnitude. In the Tapajós catchment, in EXDEF scenario, LPJmL-DGVM exhibits strong water limitations and produces nearly no evaporation of intercepted rainfall, while the bare soil evaporation increases during the wet season. Over the Xingu catchment, ET reduction starts one month later in ORCHIDEE (one month earlier in INLAND-DGVM) than in the Tapajós catchment, and seasonal variation

of △ET does not change with LPJmL-DGVM (Fig. 12d). Over the Madeira catchment, both ORCHIDEE and INLAND-DGVM simulate a small decrease in ET in the EXDEF simulations during the dry season (by up to -10 mm month$^{-1}$ in August) while LPJmL-DGVM produces no change of ET (Fig. 12b). Changes in ET component fluxes have the same signs as in the Tapajós but smaller magnitudes (compare Fig. S1a and Fig. S1b).

### 3.3.3 South-eastern catchments: uncertainties due to model structure

Deforestation-induced ET variations during the dry season are driven by soil moisture changes which limit ET from dry soils (Juárez et al., 2007; Guimberteau et al., 2014). Thus, ET and runoff variations simulated by the LSMs are strongly linked to their soil hydrology and different soil moisture parameterizations, soil depths (2 m, 3 m and 4 m in the case of ORCHIDEE, LPJmL-DGVM and INLAND-DGVM, respectively) and soil texture maps (Table 1). Looking at specific model behavior, e.g. during the dry season in the Tapajós catchment for the EXDEF scenario and CCSM3 forcing, we found that deforestation

decreases soil moisture in the upper layers in ORCHIDEE and INLAND-DGVM (down to 50 cm and 2 m, respectively) while deeper soil moisture increased in these two LSMs (Fig. 13a). These opposing changes of soil moisture in the soil profile are explained by the substitution of the deep-rooted forests by shallow-rooted pasture and crops in the two LSMs (see de Rosnay and Polcher (1998) for ORCHIDEE and Kucharik et al. (2000) for INLAND-DGVM). Short vegetation can only access water for transpiration from the near-surface layers. The resulting deforestation-induced ASO transpiration decrease is

higher with ORCHIDEE ($\sim$ 30 mm month$^{-1}$ in August) than INLAND-DGVM ($\sim$ 15 mm month$^{-1}$ in the same month). Yet, in both conditions (with or without deforestation), INLAND-DGVM simulates higher ASO transpiration (Fig. 13a). This can be explained by the higher soil water holding capacity of INLAND-DGVM which enables this LSM to carry over more water from the wet season than ORCHIDEE. This helps to sustain the evaporation during the dry season, as reported by Guimberteau et al. (2014). The simulated LAI being higher in INLAND-DGVM than in ORCHIDEE can also explain the differences between

the two LSMs in simulated ET (not shown). In contrast to both ORCHIDEE and INLAND-DGVM, transpiration with LPJmL-DGVM is strongly limited by water availability nearly all year round in southern Amazonia. As a result of this background limitation even without deforestation, under the EXDEF scenario, soil moisture in the deep layers of LPJmL-DGVM decreases only slightly all year long (by $\approx$ 10 mm, Fig. 13a) and transpiration does not change during the dry season.




### 3.3.4 South-eastern catchments: changes in soil moisture explained by GCM precipitation seasonality in presence of deforestation

The amplitudes of the seasonal cycle of precipitation are different between the GCM-forcings. In the UKMO-HadCM3 model, the seasonal amplitude is lower than in CCSM3 and PCM (compare Figs 13a and 13b). In southern Amazonia, the CCSM3 precipitation drops by 79% (–300 mm month$^{-1}$) between March (wettest month) and May (beginning of the dry season). By contrast, the precipitation drop between these two months is 60% (–180.0 mm month$^{-1}$) in UKMO-HadCM3. The influence of precipitation from the GCM-forcings on the response of soil moisture variation to deforestation depends on the LSM considered. As a result, soil moisture is lower all year long with LPJmL-DGVM and ORCHIDEE forced by UKMO-HadCM3 due to the dry condition of the soil, even if the deforestation reduces ET. The largest soil moisture decrease occurs with OR-CHIDEE from March to June, during the beginning of the dry season. Thus, the change in transpiration simulated by this LSM is highly sensitive to the difference in precipitation changes during the wet-to-dry transition period, between CCSM3 and UKMO-HadCM3.

### 3.3.5 Changes in runoff and river discharge

The increase of runoff simulated over the catchments translates into an increase of river discharge through the routing schemes of ORCHIDEE and LPJmL-DGVM. Because of the small effects of deforestation and climate change on the water budget of the entire Amazon basin, changes in river discharge simulated by the LPJmL-DGVM, which is already dry in regions affected by deforestation (see above), are negligible for all the sub-catchments (Fig. 14). The seasonal river discharge simulated by ORCHIDEE at Óbidos, the gauging station closest to the mouth of the Amazon basin, is little affected by deforestation and climate change, with just a slight shift in the discharge increase between December and May. This is explained by an increase in runoff at the end of the dry season in the south (MAD), where river discharge slightly increases between October and April.

The discharge extremes of the southern rivers (Madeira and Tapajós) are affected by deforestation (Fig. 15). For the Madeira, the deforestation in the EXDEF scenario has an opposite effect compared to the effect of climate change on the low flows. Namely, the deforestation induced increase of low flow ranges from 5.0 to 14.5%, according to different LCC scenarios and compared to a climate change induced reduction by 50% (Fig. 15a). The high-flow increase in the Madeira due to deforestation is very small (0.5%) when compared to the climate change effect (+15%). Yet, the spread of the results in high-flow increase due to climate change is significantly reduced when taking the deforestation into account, suggesting some robustness in the simulated impacts of deforestation on high flows. In the Tapajós catchment with the largest future forest area loss, high flows do not change with deforestation, while climate change alone increases them by 12% but with a large spread (Fig. 15b). In contrast, with the deforestation, low flows consistently increase from 15 to 32% in the 2090's, depending on the LCC scenario. These changes are of same magnitude as the effect of climate change (-31%) suggesting that future deforestation is likely to offset the climate change impact on the hydrology of this catchment.



## 4 Discussion and synthesis

### 4.1 Does deforestation balance or amplify the impact of climate change on the hydrology of the Amazon basin?

Although with high uncertainties, greenhouse gas-induced climate change will probably enhance the water cycle in Amazonia, increasing annual precipitation, ET and runoff by the end of the century. The three LSMs used in this study simulate an in-
crease of ET, despite the physiological (anti-transpirant) effect of increased $CO_2$ being accounted for in all of them. However, this behaviour needs to be considered with caution as it is obtained without considering atmospheric feedbacks. Considering the land-atmosphere coupling, deforestation may change precipitation recycling and thus the sign of the water balance over Amazonia (Coe et al., 2009). Consistent with Cook et al. (2012), Langerwisch et al. (2013), Guimberteau et al. (2013) and Sorribas et al. (2016), contrasted precipitation changes are projected between southern and western-northern regions. Com-
paring Figures 4b and 8b, the most pronounced decrease in ET occurs during the end of the dry season, in agreement with Lejeune et al. (2014), in regions where precipitation declines. ET decreases more than precipitation over all the south-eastern catchments (TAP and XIN), i.e., land surface processes incorporated in LSMs reduce the evaporated fraction of precipitation.

It has been suggested that a reduction in the area of Amazonian forest, such as produced by the EXDEF scenario, will push much of Amazonia into a permanently drier climate regime (Malhi et al., 2008). At annual scale, deforestation-reduced
ET only partly offsetting the positive effect of climate change on ET even in EXDEF, so that all the simulations give a net increase of runoff by the end of the century. In south-eastern Amazonia, the ~50% forest area loss in EXDEF combined with climate change leads to a consistent ET decrease which offsets positive changes of ET due to climate change alone. Over the Xingu, our projections of the hydrological budget are consistent with Panday et al. (2015), who also found opposite effects of deforestation and climate during the past 40 years using a combination of long-term observations of rainfall and discharge. Yet,
our settings of constant land-use ignore the influence of historic deforestation on ET and may result highly biased estimates of deforestation effects on ET.

The resulting increase of runoff in the Xingu catchment (+9%), with increasing deforestation in the future, is of similar order to the results of Stickler et al. (2013), who found a 10 to 12% runoff increase given 40% deforestation in this catchment. Yet, during ASO in the south-eastern catchments, deforestation amplifies the effect of climate change in reducing ET, in particular
in the south of the Tapajós catchment and in the north of the Madeira and Xingu catchments where deforested areas are the largest. Thus, deforestation contributes to the increase in runoff (+35% in TAP), and thus balances the runoff-reducing effect of climate change (-22% in the Tapajós).

### 4.2 Consequences on the extreme discharge of the southern rivers

The ET decrease and runoff increase projected for southern catchments (MAD and the TAP) by the extreme deforestation
scenario applied here (EXDEF) balances the climate change effect on low flows. Climate change alone increases the seasonal amplitude of discharge, and high-flow values. In contrast, deforestation balances this effect by reducing the risk of decrease in low flows in the Madeira and Tapajós in all LCC scenarios; this is related to the decrease of ET during the dry season. Our result for the Madeira contradicts those of Siqueira Júnior et al. (2015) who found a decrease of low flows with the hydrological





model MHD-INPE, combined with the «business-as-usual» scenario (BAU) of Soares-Filho et al. (2006) where deforestation is lower than in the EXDEF scenario. They argue that this behaviour is due to the occurrence of faster flows when deforestation is taken into account; even though this contradicts the fact that LCC scenarios are associated with reduced ET. This comparison highlights the uncertainty in the results of the effect of deforestation on hydrology, depending on whether we use LSMs or hydrological models to simulate river discharge.

### 4.3   Are the uncertainties of the model responses to deforestation and climate change greater than those from GCM projections?

Our results confirm the large climate change uncertainty, shown in previous studies (Li et al., 2006; Vera et al., 2006; Torres and Marengo, 2013), with the three GCMs giving different signs of precipitation changes, in particular in the southern water-limited regions of the Xingu and Tapajós catchments. In these catchments, the sign of the variation of ET and runoff due to climate change alone is particularly uncertain, due to the large spread in GCM rainfall projections. The impact of deforestation combined with climate change on soil moisture and ET also depends on the GCM forcing. Yet, the magnitude of the ET changes due to climate change alone is more uncertain than that induced by the deforestation in the three LSMs assessed. During the wet-to-dry transition period, the strength of the precipitation decrease driven by climate change determines the change of soil water storage due to deforestation which sustains the evaporation during the dry season. Our study emphasizes the uncertainty associated with the choice of the LSM to be paired with the GCM forcing, and the effect this choice has on the inherent uncertainties the model's parameterization (energy or water-limited) produces on estimating deforestation impacts on hydrology. Over large river basins like the Amazon, these models have the disadvantage of being rather poorly constrained in their parameterizations of both vegetation functioning (Poulter et al., 2010) and soil hydrology (Christoffersen et al., 2014). In our view, the LSM community needs to strengthen its efforts to cooperate with the soil science community to improve the representation of soil hydrological processes in their models, despite the difference in scale at which they work and the inherent small-scale variability of soil properties. Nevertheless, the uncertainty among the models is lower than among climate projections. The magnitude of the changes in soil hydrology first depends on GCM precipitation changes, in particular during the beginning of the dry season and then on the behaviour of each LSM (water versus energy-limited models).

### 5   Conclusions

The construction of new land-cover change scenarios for Amazonia indicates that, by the end of this century, the total forested area of the Amazon basin will have decreased by 7% in the best case, to 34% in the most severe scenario. The most severe forest clearing occurs in southern Amazonia where the Madeira, Xingu and Tapajós catchments experience a 50% decrease in forest area. With a multi-model approach, we found that the replacement of the forests by pasture and crops should only slightly decrease annual evapotranspiration by up to 2.5% and enhance runoff by up to 3.0%, for the most severe scenario of the Amazon basin.





The south-eastern catchments, however, are more vulnerable at the end of the dry season. Compared to forest, crops and pastures fail to sustain their evaporation in a high drought stress context. Given the combination of decreased rainfall due to future climate change and the large forest area loss, evapotranspiration may drop by -10 and -12% in the Xingu and Tapajós catchments, respectively, deforestation amplifying the decrease of ET due to climate change. These results strongly depend on

the land surface model used, they vary with the soil hydrology parameterizations; but they also depend on the climate model forcing data. In contrast, by enhancing the runoff, the deforestation balances the negative effect of climate change on runoff in these catchments. As a result, the deforestation in the most intensive scenario balances the risk of decrease in low flows of the Tapajós due to climate change by the end of the century. Our results in the Tapajós catchment emphasize the impact of deforestation combined with climate change on hydrological extremes. The deforestation leads to a 32% increase in low flows

by the end of the century, which offsets the opposite impact of climate change.

Biosphere-atmosphere interactions, not accounted for in our study, are also crucial in estimating the progress of forest dieback, whereby forest is replaced by savanna vegetation. During the end of the dry season, we found a strong reduction of ET in south-eastern Amazonia. Evaporation at this time of year provides a critical source of water vapour for precipitation and lower ET can delay the onset of the wet season (Fu and Li, 2004) and reduce the water recycling during this period (Lima et al., 2014).

We need to pay careful attention to the intensification and lengthening of droughts during this century; a phenomenon that is commonly predicted by the GCMs for southern Amazonia (Boisier et al., 2015). Whatever its cause, our results emphasize the need to include the deforestation process in climate change simulations. Deforestation has the potential to mask (or unmask) the effects of climate change on surface hydrology.

**Competing interests**

The authors declare that they have no conflict of interest.

*Acknowledgements.* This work was financially supported by the EU-FP7 AMAZALERT (Raising the alert about critical feedbacks between climate and long-term land-use change in the Amazon) project (Grant Agreement No. 282664) and the European Research Council Synergy grant ERC-2013-SyG-610028 IMBALANCE-P. We acknowledge the SO HYBAM team who provided their river flow data sets for the Amazon basin (http://www.ore-hybam.org). Simulations with ORCHIDEE were performed using computational facilities of the Institut du

Développement et des Ressources en Informatique Scientifique (IDRIS, CNRS, France). Grateful acknowledgement for proofreading and correcting the English edition goes to John Gash.



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





| Model | Institute | Reference | Model setup |
|---|---|---|---|
| ORCHIDEE[1] | IPSL, Paris, France | Krinner et al. (2005) | River routing including floodplains/swamps, FAO soil texture map |
| INLAND-DGVM[2] | INPE, Sao Jose dos Campos, Brazil | Foley et al. (1996) Kucharik et al. (2000) | Fire emissions, soil texture map |
| LPJmL-DGVM[3] | PIK, Potsdam, Germany | Sitch et al., 2003 | Fire emissions, river routing, FAO soil texture map |

[1] ORganising Carbon and Hydrology In Dynamic EcosystEms
[2] INtegrated model of LAND surface processes
[3] Lund Potsdam Jena managed Land model

**Table 1.** Models used in this study

| Name of the simulation | Scenarios | |
|---|---|---|
| | CC | LCC |
| HIST | no | no |
| CCSM3 NODEF | yes | no |
| CCSM3 LODEF | yes | yes |
| CCSM3 HIDEF | yes | yes |
| CCSM3 EXDEF | yes | yes |
| UKMO-HadCM3 NODEF | yes | no |
| UKMO-HadCM3 LODEF | yes | yes |
| UKMO-HadCM3 HIDEF | yes | yes |
| UKMO-HadCM3 EXDEF | yes | yes |
| PCM NODEF | yes | no |
| PCM LODEF | yes | yes |
| PCM HIDEF | yes | yes |
| PCM EXDEF | yes | yes |

**Table 2.** List of the different simulations performed with the three LSMs (ORCHIDEE, INLAND-DGVM and LPJmL-DGVM) with or without climate change (CC) and land-cover change (LCC).

| Institutes, country | Model | | Resolution | References |
|---|---|---|---|---|
| | Name | Acronym | (lat x lon) | |
| National Center for AtmosphericResearch (NCAR), USA | Community Climate System Model | CCSM3 | ~1.4° x 1.4° | Bonan et al. (2002) |
| | Parallel Climate Model | PCM | ~2.8° x 2.8° | Washington et al. (2000) |
| Hadley Centre for Climate Prediction and Research / Met Office, UK | Hadley Centre Coupled Model | UKMO-HadCM3 | ~2.5° x 3.75° | Gordon et al. (2000) |

**Table 3.** List of the GCMs participating in CMIP3 used in this study with their approximate atmospheric horizontal resolution.





| Qualitative scenarios | | Quantification of deforestation rates (spatially explicit projections until 2100) | | | |
|---|---|---|---|---|---|
| Name | Brief storyline | Name | Brazilian Amazon (Aguiar et al., 2016) | Bolivian Amazon (Tejada et al., 2015) | Other countries (Eupen et al., 2014) |
| A-Sustainability | "Zero" deforestation scenario. Sustainable land use, protected areas, indigenous territories, restrained construction of new roads. | LODEF | Annual rate decreasing to 3,900 km$^2$ yr$^{-1}$ until 2020, and then to 1,000 km$^2$ yr$^{-1}$ until 2025, and then stabilizing until 2100. | Trend of 2005–2008 until 2013, then decrease by 50%. | Same as HIDEF |
| C-Fragmentation | Return of high deforestation rates | HIDEF | Annual rate increasing to 15,000 km$^2$ yr$^{-1}$ until 2020 and stabilizing until 2100. | Total deforested area reaches 13 million in 2025 ha, then replicates the 2005–2008 annual rate. | For each country, projected according to historical trends. |
| | | EXDEF | Annual rate increasing to 19,500 km$^2$ yr$^{-1}$ (1996-2005 historical rate) until 2020 and stabilizing until 2100. | Same as HIDEF | Same as HIDEF |

**Table 4.** LCC scenarios used in this study

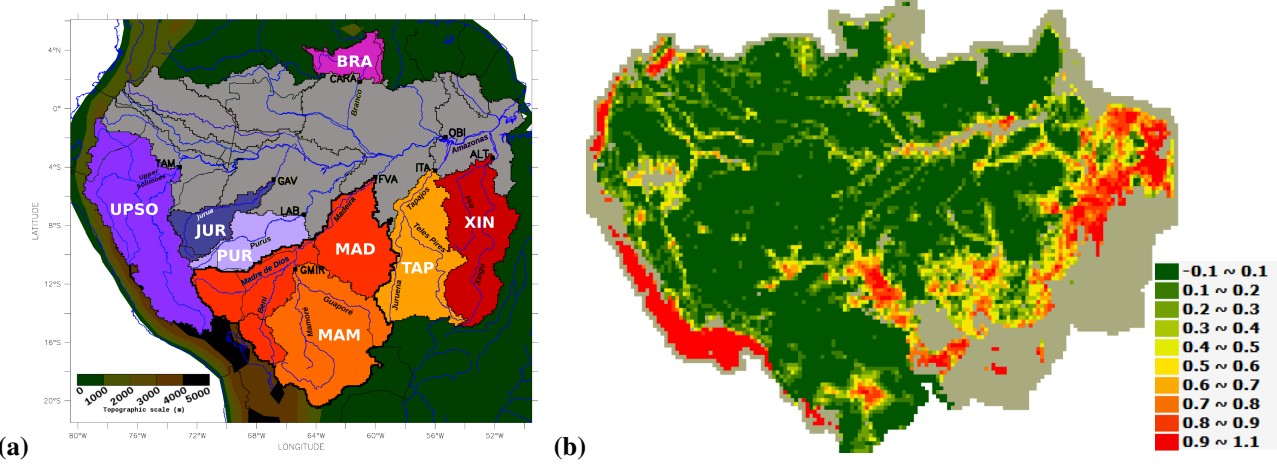

**Figure 1. (a)** Amazon catchments (name in white, the abbreviations are indicated in Table S1) and the main rivers (adapted from Guimberteau et al., 2012). Localization of the main SO HYBAM gauging stations (the abbreviations are indicated in Table S1). The bold black line delineates the Madeira catchment. Colour is used to distinguish the southern (red), western (purple) and northern sub-catchment (pink). Topographic scale is indicated. **(b)** Percentage of deforestation in each 25 x 25 km$^2$ in 2005 (observed data, Aguiar et al., 2016).



**Figure 2.** Flow chart methodological approach for historical and future simulation processes (CC = climate change, LCC = land-cover change). Acronyms of the LCC scenarios are explained in Table 4.





**Figure 3.** Changes in mean annual (mm yr$^{-1}$) and ASO values (mm month$^{-1}$) of (**a** and **b**) precipitation, (**c** and **d**) ET and (**e** and **f**) runoff due to climate change only, for the end of the century, over the Amazon basin and eight of its sub-basins (the abbreviations of the sub-basins are indicated in Table S1). Each box plot corresponds to the interquartile range (IQR, distance between the 25$^{th}$ and the 75$^{th}$ percentiles) within each basin indicating the spread of the 3 GCM-forcings (for ΔP) and 3 GCM-forcings x 3 LSMs (for ΔET and ΔR) results (see Figure 1 for colour code). For a given box plot, the black points denote the mean value over the basin, the whiskers extend from the minimum value to the maximum one and the numbers above the box plot indicate the mean relative differences over the basin (%).





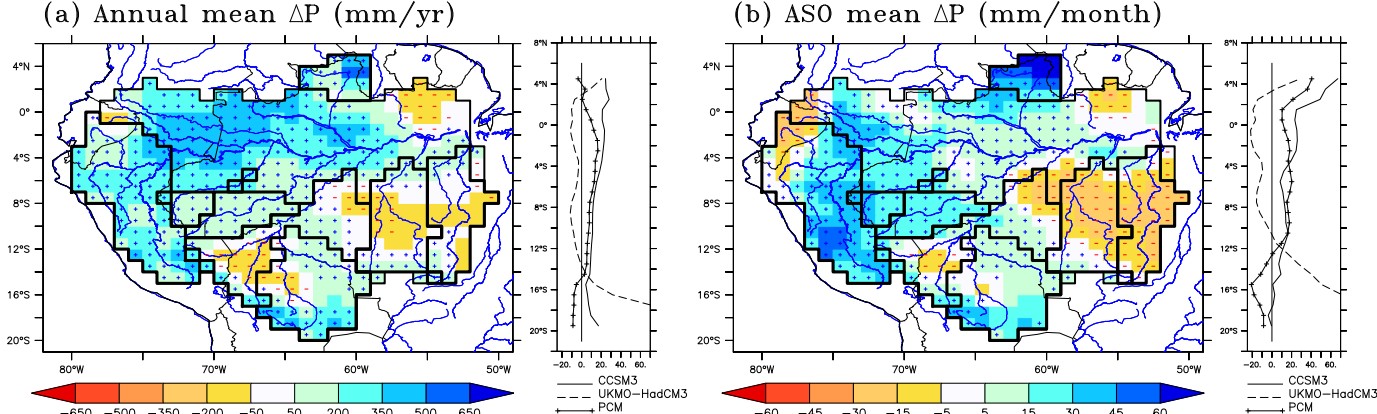

**Figure 4. Maps**: Spatial change in **(a)** annual ($\mathrm{mm\,yr^{-1}}$) and **(b)** ASO precipitation ($\mathrm{mm\,month^{-1}}$) due to climate change (mean of the three GCM-forcings), for the end of the century. The symbols indicate that more than two GCM-forcings out of three give a precipitation increase (+) or decrease (-) on the grid cell. The black lines delineate the Amazon basin and the sub-basins. **Plots**: Corresponding zonal mean of relative changes in precipitation (%) from the three GCM-forcings.

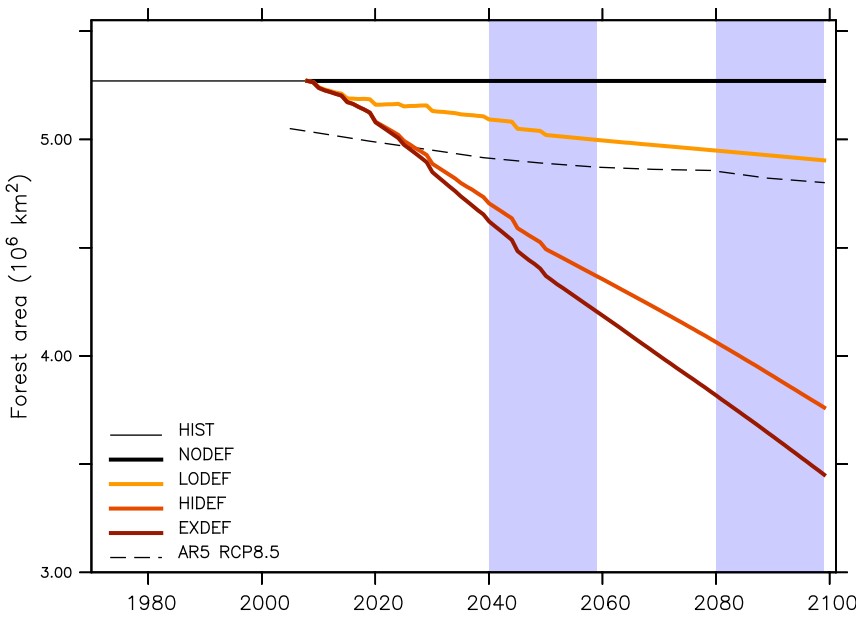

**Figure 5.** Interannual variation of forest area ($10^6\,\mathrm{km}^2$) over the Amazon basin, according to the NODEF scenarios, the three LCC scenarios and the SSP of the AR5 RCP8.5 scenario over 2009-2099. The blue bands indicate the two future periods selected for this study.





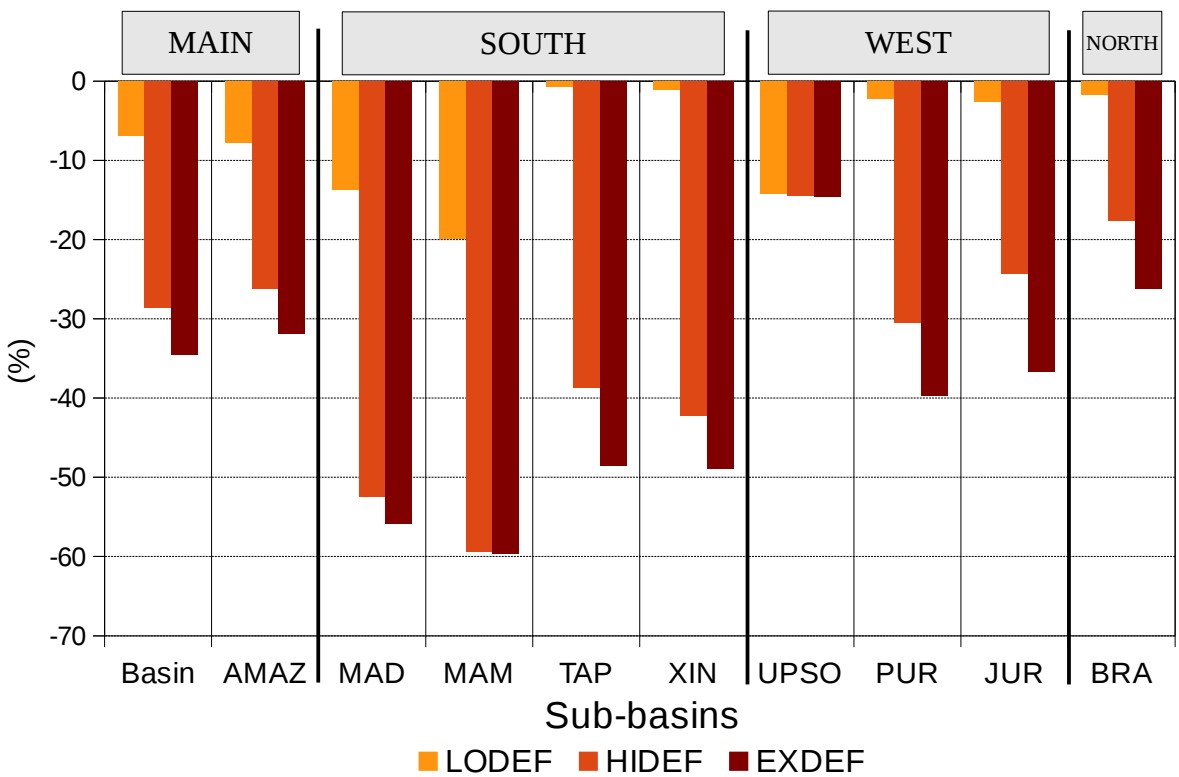

**Figure 6.** Forest area decrease (%) over the different sub-basins of the Amazon basin between each of the three LCC scenarios in 2099 and the NODEF scenario in 2009 (the abbreviations of the sub-basins are indicated in Sect. 2.1 and Table S1).







**Figure 7.** Decrease of forest fraction for the three LCC scenarios (for the two time periods) compared with the NODEF scenario in 2009 over the Amazon basin. Grey colour indicates no change of forest fraction.





**Figure 8. Maps**: Spatial change in **(a,c)** annual (mm yr$^{-1}$) and **(b,d)** ASO ET (mm month$^{-1}$) due to **(a,b)** climate change (mean of the three GCM-forcings) and **(c,d)** deforestation combined with climate change (EXDEF), for the end of the century. The symbols indicate that more than six simulations out of nine (3 GCM-forcings x 3 LSMs) give an increase (+) or a decrease (-) of ET on the grid cell. The black lines delineate the Amazon basin and the sub-basins. **Plots**: Corresponding zonal mean of relative changes in ET (%) from each of the nine simulations.





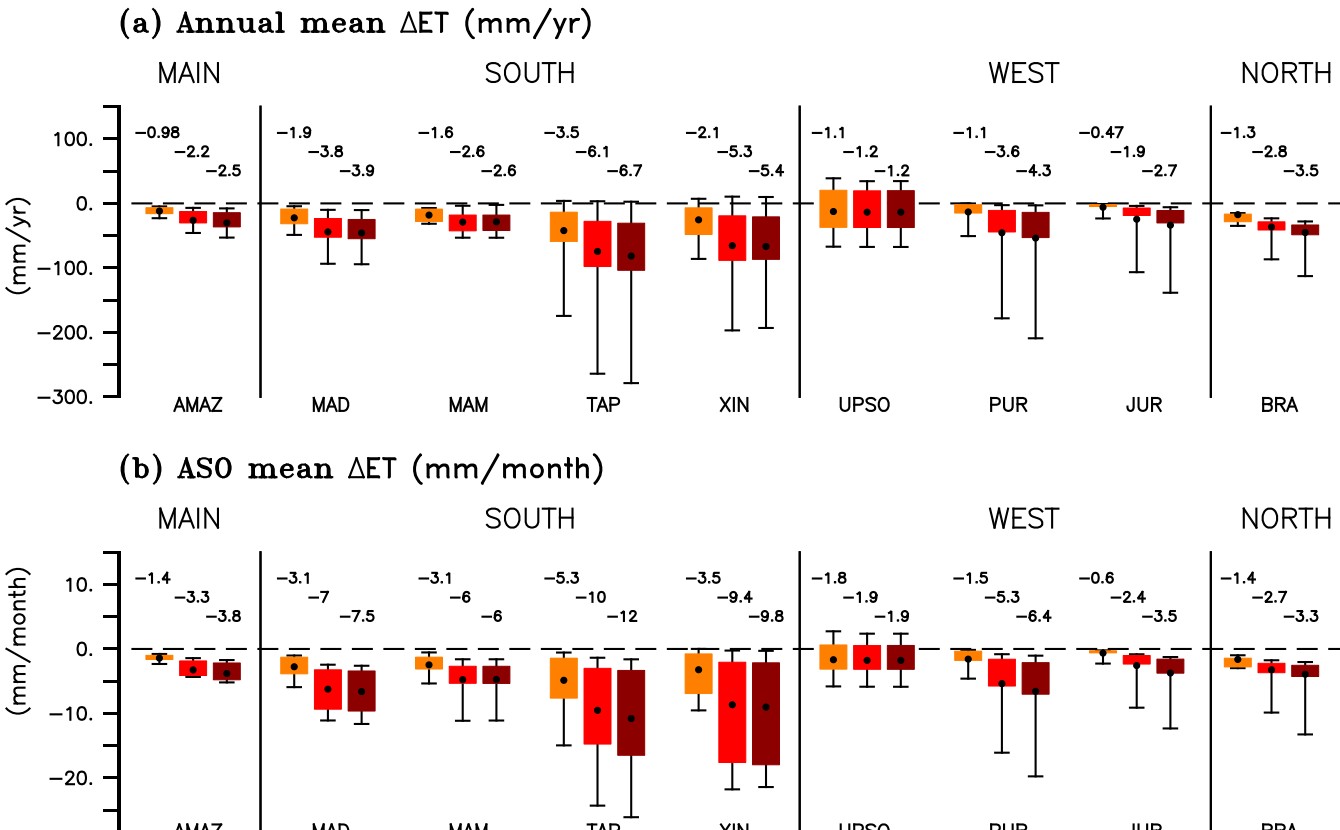

**Figure 9. (a)** Mean annual ($mm\,yr^{-1}$) and **(b)** ASO values ($mm\,month^{-1}$) of ET changes due to deforestation and assuming future climate change, for the end of the century, over the Amazon basin and eight of its sub-basins (the abbreviations of the basins are indicated in Table S1). Each box plot corresponds to the interquartile range (IQR, distance between the 25th and the 75th percentiles) within each basin indicating the spread of the 3 GCM-forcings x 3 LSMs results for one LCC scenario (see Figure 6 for colour code). For a given box plot, the black points denote the mean value over the basin, the whiskers extend from the minimum value to the maximum one and the numbers above the box plot indicate the mean relative differences over the basin (%).





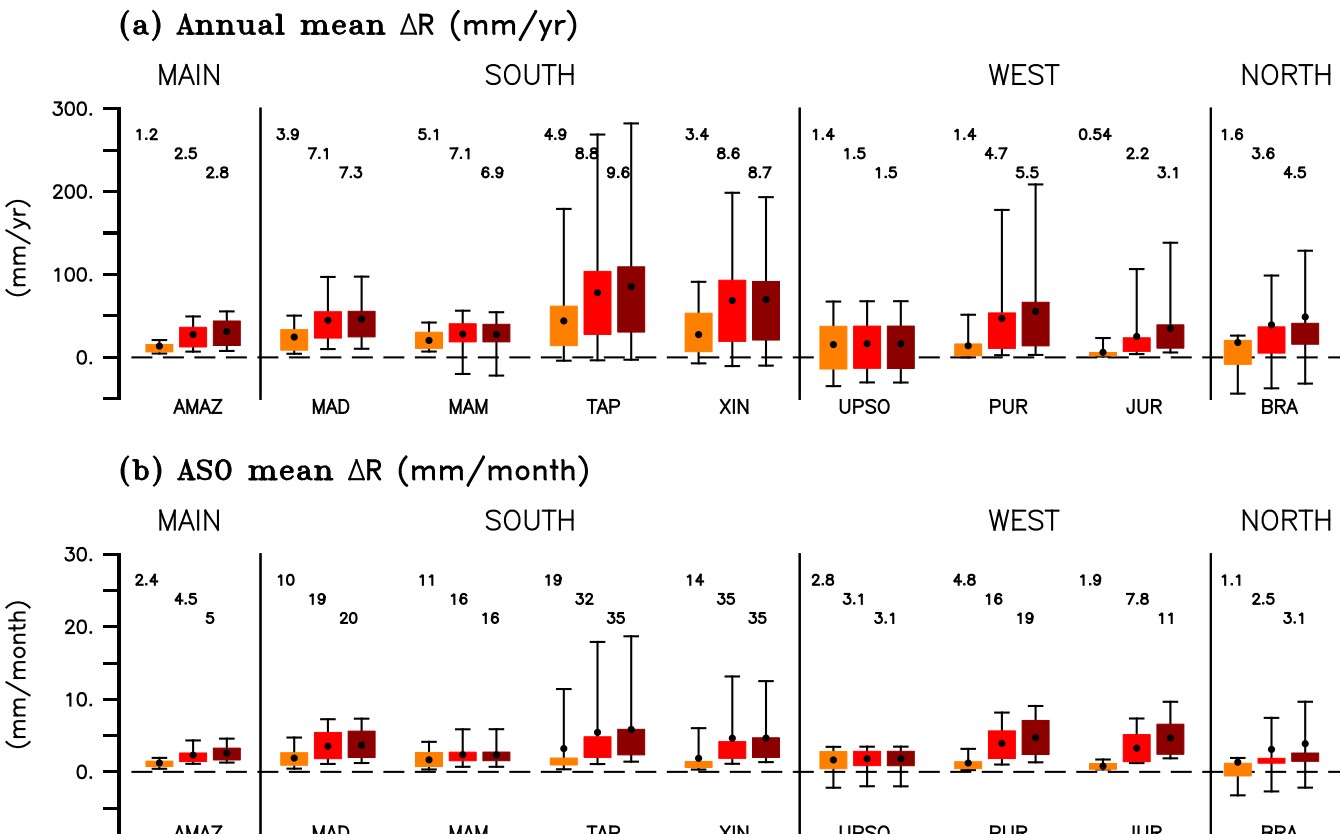

**Figure 10.** As in Figure 9 but for runoff.





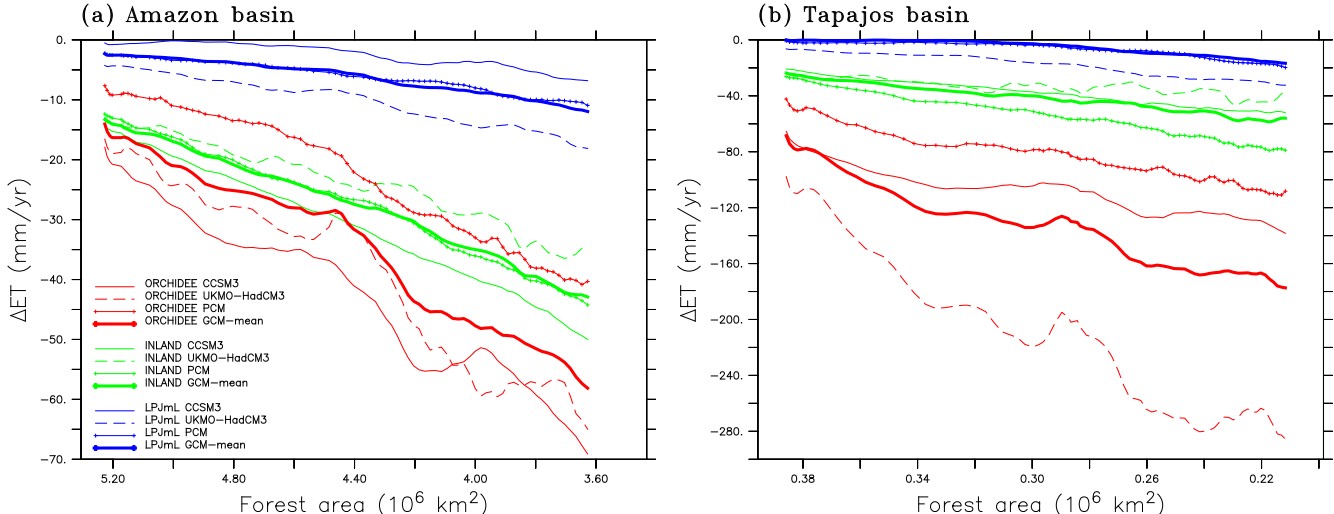

**Figure 11.** ET changes (mm yr$^{-1}$) over 2009-2100 (1-year running mean) as a function of tree area decrease ($10^6$ km$^2$) from scenario EXDEF within the **(a)** Amazon and **(b)** Tapajós basins.





**Figure 12.** Seasonal change in ET (mm month$^{-1}$) due to deforestation combined with climate change (EXDEF) simulated by the three LSMs over the Amazon and the sub-basins, averaged over the two future periods. For a given LSM and period, the shaded area defines the envelope enclosing the range with plausible climate futures.





**Figure 13.** Impact of deforestation combined with climate change on ET (mm month$^{-1}$), transpiration (mm month$^{-1}$), runoff (mm month$^{-1}$) and soil moisture (mm) over the Tapajós catchment for the three LSMs, for two different end of century climates: **(a)** CCSM3 and **(b)** UKMO-HadCM3. **Top panels**: seasonal cycle of precipitation (mm d$^{-1}$) with colour bars and seasonal cycles of ET, transpiration and runoff with plain/dashed lines for NODEF/EXDEF LCC scenarios. **Bottom panels**: Corresponding $\Delta$soil moisture (mm) due to deforestation combined with climate change. Results for PCM forcing are similar to those for CCSM3 forcing (not shown).





**Figure 14.** Seasonal river discharge (m$^3$ s$^{-1}$) simulated by ORCHIDEE and LPJmL-DGVM from HIST (averaged over 1970-2008) and from NODEF and EXDEF LCC scenarios (mean of the three GCM-forcings for each scenario, averaged over 2080-2099) at the gauging stations over the Amazon sub-basins. The results from HIST simulations are compared with the observations from the SO HYBAM (averaged over 1970-2008).





**Figure 15.** Relative change (%) of the first deciles (i.e., low flow, left panels) and the last deciles (i.e., high flow, right panels) of river discharge due to climate change (grey) and deforestation combined to climate change (three LCC scenarios) of **(a)** the Madeira (at FVA) and **(b)** the Tapajós (at ITA), for the middle (green) and the end (red) of the century. The changes are simulated by the ensemble of six simulations (2 LSMs x 3 GCM-forcings).