# Peer review of "Impacts of future deforestation and climate change on the hydrology of the Amazon basin: a multi-model analysis with a new set of land-cover change scenarios"

_Hydrology and Earth System Sciences, 2016_

## Referee Comment (RC1) · Anonymous Referee #1 · 14 Sep 2016

The study of Guimberteau et al. applied three land surface models forced with three different GCM's in combination with different land use change scenarios to the Amazone basin. The authors show that due to climate change alone temperature increases, as well as precipitation and evaporation. Under the different land use change scenarios, transpiration decreases and runoff increases. In general, the article is very interesting, but some clarifications might be needed.

General:

1) One point of confusion is introduced in Line 4-5 from page 6, Section 2.3. As I

understand correctly, all the percentages and differences for the scenarios with future land use change (LODEF, HIDEF and EXDEF) are relative to a future scenario with climate change only (NODEF). This would mean that all the percentages and differences mentioned in, for example, Section 3.3 and Figures 9 and 10, are referring to the difference between two future scenarios at the same point in time (the year 2099). In this way, figures like Figure 15 are not very fair, as apples and oranges are compared, with a different benchmark. It may be more interesting to compare all the results with the same benchmark (thus, HIST and the year 2009). Now, it becomes hard to answer, for example, the question posed in the title of Section 4.1, as the percentages only reflect the isolated effect of deforestation. Thus, it can only be concluded that evaporation with a forest land cover is higher compared to non-forest. Considering this, I would like to point at other land use change experiments, which in general show that water yields increase after deforestation, in line with the findings presented here (e.g. Hornbeck et al., 2014; Rothacher, 1970; Swift and Swank, 1981). Overviews of these experiments are given by, for example, Bosch and Hewlett (1982), Andréassian (2004) and Brown et al. (2005). I may misunderstand the percentages used in the manuscript, but, in that case, please try to clarify what you are actually comparing and try to state clearly what the percentages are relative to.

2) I find the discussion in section 3.3.3 very interesting. Nevertheless, the modelled and observed river discharges in Figure 14, may add several discussion points. It can be noted that Orchidee is much closer to the observations (HIST-scenario) compared to LPJml (for subfigure MAIN, AMAZ). In this way, it can be argued that Orchidee may better represent the current processes, and may (but not necessarily) also better reflect what happens in the future scenarios. The opposite reasoning may also hold. Why trust a model that is not able to capture the historical series? Anyway, it may be interesting to reflect on these issues.

3) Throughout the paper, the term evapotranspiration (ET) is used, whereas the more general term 'evaporation' or 'total evaporation' may be more clear. I would like to

refer to Savenije (2004) for some arguments to not use the term evapotranspiration as well. Briefly, transpiration is a rather different process compared to, for example, interception evaporation. Especially with regard to deforestation, it is important to make this distinction, as it is probably transpiration that decreases.

Detailed comments:

P4.L30. What do you mean with "business-as-usual"? The current situation?

P12.L1. Idem

P23.Fig3. Define the abbreviation of ASO (now only later in the text)

P33.Fig.14. Why is the third model not shown?

Concluding, I think the work is interesting and worth publishing in HESS. Nevertheless, some efforts will be needed to clarify the issues stated above.

References

Andréassian, V.: Waters and forests: from historical controversy to scientific debate, Journal of Hydrology, 291, 1-27, http://dx.doi.org/10.1016/j.jhydrol.2003.12.015, 2004. Bosch, J. M., and Hewlett, J. D.: A review of catchment experiments to determine the effect of vegetation changes on water yield and evapotranspiration, Journal of Hydrology, 55, 3-23, http://dx.doi.org/10.1016/0022-1694(82)90117-2, 1982.

Brown, A. E., Zhang, L., McMahon, T. A., Western, A. W., and Vertessy, R. A.: A review of paired catchment studies for determining changes in water yield resulting from alterations in vegetation, Journal of Hydrology, 310, 28-61, http://dx.doi.org/10.1016/j.jhydrol.2004.12.010, 2005.

Hornbeck, J. W., Eagar, C., Bailey, A., and Campbell, J. L.: Comparisons with results from the Hubbard Brook Experimental Forest in the Northern Appalachians, Long-Term Response of a Forest Watershed Ecosystem: Clearcutting in the Southern Appalachians, 213, 2014.

Rothacher, J.: Increases in Water Yield Following Clear-Cut Logging in the Pacific Northwest, Water Resources Research, 6, 653-658, 10.1029/WR006i002p00653, 1970. Savenije, H. H. G.: The importance of interception and why we should delete the term evapotranspiration from our vocabulary, Hydrological Processes, 18, 1507-1511, 10.1002/hyp.5563, 2004.

Swift, L. W., and Swank, W. T.: Long term responses of streamflow following clearcutting and regrowth / Réactions à long terme du débit des cours d'eau après coupe et repeuplement, Hydrological Sciences Bulletin, 26, 245-256, 10.1080/02626668109490884, 1981.
* * *

---

## Referee Comment (RC2) · Anonymous Referee #2 · 14 Sep 2016

Scientific significance:

This manuscript uses 3 Global Climate Models (GCMs), 3 Land Surface Models (LSMs), and 3 Land Cover Change (LCC) scenarios to investigate the impacts of future deforestation and climate change on the hydrology of the Amazon Basin. The scientific questions posed, i.e. What is the direction of projected change; how is hydrologic change apportioned between climate change and land cover change forcings; and what are the relative uncertainlties introduced by different GCMs, LSMs and LCC scenarios are of interest to the international hydrologic sciences community and within

the scope of HESS.

The techniques and tools utilized in the study are standard: Use of publically available (and slighlty outdated) CMIP3 models, use of existing LSM models, and development of a suite of Land Cover Scenarios using a combination of a participatory approach with existing land use models. No new data is collected and, as discussed below, no calibration or validation of the LSM models against existing hydrologic data is presented. As such the paper presents no new methodological contributions; its contribution is primarily in the integration of existing tools and synthesis of resulting predictions to answer important questions.

One of the major scientific limitations of the study, the omission of any feedbacks between land cover change and climate change, is acknowledged in the first sentence of the abstract and the last paragraph of the conclusions. While rectifying this omission is likely beyond the scope of this study or this team of researchers, it does call into question whether the study's conclusions will hold up under the likely scenario that changes in evapotranspiration as a result of amazon deforestation change the regional climate. Nevertheless, with a more rigorous validation of LSM predictions of historical hydrology, a more quantitative partitioning of sources of prediction uncertainty among GCMs, LSMs, and LCC scenarios , and a stronger synthesis of results into new insights or actionable information I believe this paper would be of interest to the HESS readership. Detailed suggestions are given below.

Scientific quality:

While the scientific approach seems valid, insufficient details are given on the modeling methodology, for example the following details are not clear:

1) More details on the GCM bias correction and statistical downscaling methods should be provided. Since the authors are evidently using others' results the methodology descriptions do not need to be exhaustive but the methods used in the referenced papers should be identified so readers do not have to look up the previous work.

2)The LSMs are not adequately or consistently described in the manuscript or supplemental material. It is not clear how the important hydrologic processes are represented, or how they are parameterized, in the three LSMs. Details regarding the spatial and temporal resolutions of the models should be presented in Table 2, along with much more detail regarding how particular hydrologic processes (evaporation, transpiration, unsaturated flow, groundwater flow, overland flow, river routing) are simulated in the models. The description of the models in the supplementary material is qualitative rather than quantitative and focuses most strongly on vegetative processes.

3)No evidence is given that the LSMs are able to adequately simulate historic ET and river fluxes in the study region. The supplemental materials broadly states that two of the three models (LPJmL-DGVM and ORICHIDEE) have been widely tested, but a comparative summary of the three models predictions during the historic period should be presented. Historical rivers flows are included in Figure 14 but no attempt is made to attribute errors in predictions or discuss the relative magnitude of errors among LSMs. Section 3.3.3 discusses differences in LSM predictions in a qualitative way but without knowing specifics of how the processes are simulated in the different models it is difficult to generalize the results beyond this modeling exercise. Since the goal of this manuscript is to understand hydrologic change it is important to show that the hydrologic models make credible predictions of ET, soil moisture, groundwater levels, river flow during the historical time period, and to understand how specific differences in hydrologic representation among the models lead to differences in predictions.

4)The description of the development of the deforestation scenarios is somewhat confusing. I am not a land use change modeler but it is not clear to me how the results from the participatory process in certain parts of the study region were extrapolated and incorporated into the LuccME and CLUE models, or why two different land use change models were required. The manuscript states (e.g. line 21 p 5 and line 34 p 5 ) that maps were used to "calibrate and validate the deforestation model", and scenarios were "translated into model parameters" but no details are given on the methods,

parameter values or prediction accuracies. I am left wondering how good the land use change models are, and how much the precise spatial distribution of deforestation shown in figure 7 matters, versus a more generic uniform decrease in forest area across the domain. If the actual locations of increased deforestation make a difference to predictions of future hydrologic change this would be interesting.

The authors present a wide range of results showing how future projections differ based on GCM, LSM and LCC scenario. However it is unclear at the end of my reading of the manuscript what are the dominant drivers of these differences, or what new insights or actionable information has been generated from this study. It would be useful if the authors could synthesize their results to quantitatively apportion uncertainty for various hydrologic predictions among the three sources (GCM, LSM, LCC). It might also be interesting to weight the ensemble of future projections based on historical reliability of the GCMs and LSMs and possibly the convergence of their future predictions (see e.g. reliability ensemble averaging (e.g. Giorgi and Mearns, Geophysical Research Letters, 2003; Asefa and Adams, Regional Climate Change 2013).

Presentation quality:

The manuscript is generally well-written, concise and well-structured. The figures are generally of good quality, however many of the labels and legends are difficult to read (e.g. Fig 1a, Fig 4 , Fig 8, Fig 13 ). In addition the figures are quite numerous. It would be helpful if the figures could be reduced and their content synthesized to more succinctly present the study's major findings and conclusions.

---

## Referee Comment (RC3) · T Roy (Referee) · 19 Sep 2016

Opinion:

This study explores how climate change and deforestation would impact the hydrology of the Amazon basin for this century, using three land surface models, three general circulation model simulations, and three (new) land cover change scenarios. The topic is quite interesting and relevant. Overall, the manuscript is well-written and well-structured. I recommend publishing this manuscript in HESS, pending some minor revisions.

[Figure]

Please see the comments below:

Major Comments:

[1] Use of the new LCC scenarios is novel in your study. Therefore, it is important to provide some literature review indicating how different your results are compared to the previous studies. You discuss Siqueira Júnior et al. (2015) in Page 11 Line 32. I suggest you provide this type of examples more often so that the manuscript is well set within the context of the existing literature.

[2] Include a discussion on why you selected the three models. I have seen that you have provided details about individual models in the supplementary materials, however, a discussion is warranted on why these three particular models were selected. How are the models different from each other and why does that difference matter? Discuss why you think that multi-model approach is better than selecting the single best model (based on historical simulation performance).

[3] Are the LSMs calibrated? It is really difficult to trust a model if it is not calibrated and evaluated. As you know, model outcomes are subject to vary (often significantly) if the parameter values are changed, given that the model structure is fixed. So, model calibration/evaluation is crucial for any model simulation-based studies. Having said that, I understand that calibrating three LSMs might be difficult. However, you should at least show how consistent the model simulations are. For this study, it would be essential to compare the historical simulations of discharge and ET from all three models against the observed data. You show historical discharge in Fig. 14, however, one of the three models is missing. I suggest you expand the discussion in Section 2.1 to address the model-related issues stated above, and add a new sub-section in Section 3 (Results) to show the historical simulation performance of the three models.

Minor Comments:

[1] Page 2 Line 24: Please add citations.

[2] Page 2 Line 27: Please add citations.

[3] Page 3 Line 16: Please state what you have done more clearly. Whether you have used a 3D matrix or not is probably not that important in the introduction section. I suggest you avoid this type of technical detailing in the introduction section and focus on them more in the 'Materials and Methods' section.

[4] Page 6 Line 2: Any specific reason for selecting these two periods?

[5] Page 8 Line 18: I see a huge difference in between-LSM ET simulations. I wonder how much of this variability is attributed to improper model calibration. Any comments?

[6] Page 31 Fig 12: How is the 'range' defined here?

[7] Page 33 Fig 14: Why are there only two models?

Figure Related Comments:

[1] Fig 1a: The axis labels and ticks are not very clear.

[2] Fig 8: The plot legends are too small. Maybe use one set of legends instead of four with a larger font?

[3] Fig 13: Maybe change the color for transpiration?

---

## Author Comment (AC1) · 28 Nov 2016

In addition to our answers to the reviewers, we re-wrote a new version of the manuscript which is available in Supplement. In this new version, the additional text is in blue and the deleted one in red.

REVIEWER

1) One point of confusion is introduced in Line 4-5 from page 6, Section 2.3. As I understand correctly, all the percentages and differences for the scenarios with future land use change (LODEF, HIDEF and EXDEF) are relative to a future scenario with climate change only (NODEF). This would mean that all the percentages and differences mentioned in, for example, Section 3.3 and Figures 9 and 10, are referring to the difference between two future scenarios at the same point in time (the year 2099). In this way, figures like Figure 15 are not very fair, as apples and oranges are compared, with a different benchmark. It may be more interesting to compare all the results with the same benchmark (thus, HIST and the year 2009).

AUTHORS

We agree with that. We recomputed all the relative difference using the same reference both against future land cover change and future climate change (HIST 2009) and we clarified this in the section 2.4 of the new version of the manuscript. The results of the percentages of change do not change a lot, except for the low-flow decrease which is less pronounced (Figure 13 of the new version of the manuscript) than with the previous computation with future extension of deforestation (as expected future deforestation already reduced low flow runoff making the % change lower when this reference was used).
* * *
REVIEWER

Now, it becomes hard to answer, for example, the question posed in the title of Section 4.1, as the percentages only reflect the isolated effect of deforestation. Thus, it can only be concluded that evaporation with a forest land cover is higher compared to non-forest. Considering this, I would like to point at other land use change experiments, which in general show that water yields increase after deforestation, in line with the findings presented here (e.g. Hornbeck et al., 2014; Rothacher, 1970; Swift and Swank, 1981). Overviews of these experiments are given by, for example, Bosch and Hewlett (1982), Andréassian (2004) and Brown et al. (2005).

AUTHORS

Thank you for these references. We have cited Hornbeck et al., 2014 and Rothacher, 1970 's results in section 4.1. in discussion.

—————————————

REVIEWER

2) I find the discussion in section 3.3.3 very interesting. Nevertheless, the modelled and observed river discharges in Figure 14, may add several discussion points. It can be noted that Orchidee is much closer to the observations (HIST-scenario) compared to LPJml (for subfigure MAIN, AMAZ). In this way, it can be argued that Orchidee may better represent the current processes, and may (but not necessarily) also better reflect what happens in the future scenarios. The opposite reasoning may also hold. Why trust a model that is not able to capture the historical series? Anyway, it may be interesting to reflect on these issues.

AUTHORS

The point [3] raised by the reviewer 3 concerning the calibration of the LSMs in historical period, is related to your question. The LSMs were not calibrated using a detailed adjustment of hydrological and water routing parameters. The best model in present time may not give necessary more trustable results in a climatic change perspective, as reported by e.g. Habets et al. (2013). In addition, dominant hydrological processes (and sources of uncertainty from each process) can change between present and future. With the CLSM model, Magand et al. (2015) found that the parameters controlling soil moisture had more influence in the future than in present, for instance. Under climatic change conditions, they found that the dominant process was no longer related to snow but rather to evapotranspiration model equations. Thus, that is why we did not trust any specific LSM more than others based on its performance for present day, even if ORCHIDEE does better for present-day, and we preferred to adopt a multi-model approach in our paper, recommended by Knutti (2010). Nevertheless, in the new Table S2 in Supplementary Material of the new manuscript, we give now the results of comparison (Relative bias, Correlation and NRMSE) between ET and runoff simulated by the three models in present time and the observations. For ET comparison, we used the machine-learning FLUXNET product (Jung et al., 2010) itself uncertain for the Amazon basin because given the small number of flux tower measurements available. The evaluation of historical ET and runoff simulated by the different models over the Amazon basin can be also found in the literature:

- for ORCHIDEE: Guimberteau et al. (2012, 2014)

- for LPJmL-DGVM: Langerwisch et al. (2013)

- for INLAND-DGVM: Dias et al. (2015), Lyra et al. (2016)

We introduced Table S2 and cited these references in section 2.1. of the new manuscript.

References cited:

- Dias L. C. P., Macedo M., Marcia N., Costa M. H., Coe M. T. and Neill C. (2015) Effects of land cover change on evapotranspiration and streamflow of small catchments in the Upper Xingu River Basin, Central Brazil, J. Hydrol.: Reg. Stud., 4, 108-122

- Guimberteau M., Drapeau G., Ronchail J., Sultan B., Polcher J., Martinez J. M., Prigent C., Guyot J. L., Cochonneau G., Espinoza J. C., Filizola N., Fraizy P., Lavado W., De Oliveira E., Pombosa R., Noriega L. and Vauchel, P. (2012) Discharge simulation in the sub-basins of the Amazon using ORCHIDEE forced by new datasets, Hydrol. Earth Syst. Sc., 16, 911-935

- Guimberteau M., Ducharne A., Ciais P., Boisier J.-P., Peng S., De Weirdt M. and Verbeeck H. (2014) Testing conceptual and physically based soil hydrology schemes against observations for the Amazon Basin, Geosci. Model Dev., 7, 1115-1136

- Habets F., Boé J., Déqué M., Ducharne A., Gascoin S., Hachour A., Martin E., Pagé C., Sauquet E., Terray L., Thiéry D., Oudin L. and Viennot P. (2013) Impact of climate

change on the hydrogeology of two basins in northern France, Clim. Change, 121, 771-785

- Knutti, R. (2010) The end of model democracy?, Climatic Change, 102:395. doi:10.1007/s10584-010-9800-2

- Jung, M.; Reichstein, M.; Ciais, P.; Seneviratne, S.; Sheffield, J.; Goulden, M.; Bonan, G.; Cescatti, A.; Chen, J.; De Jeu, R.; Johannes Dolman, A.; Eugster, W.; Gerten, D.; Gianelle, D.; Gobron, N.; Heinke, J.; Kimball, J.; Law, B. E.; Montagnani, L.; Mu, Q.; Mueller, B.; Oleson, K.; Papale, D.; Richardson, A. D.; Roupsard, O.; Running, S.; Tomelleri, E.; Viovy, N.; Weber, U.; Williams, C.; Wood, E.; Zaehle, S. & Zhang, K. Recent decline in the global land evapotranspiration trend due to limited moisture supply, Nature, 2010, 467, 951-954

- Langerwisch F., Rost S., Gerten D., Poulter B., Rammig A. and Cramer W. (2013) Potential effects of climate change on inundation patterns in the Amazon Basin, Hydrol. Earth Syst. Sc., 17, 2247-2262

- Lyra A. d. A., Chou S. C. and Sampaio G. d. O. (2016) Sensitivity of the Amazon biome to high resolution climate change projections, Acta Amazon., 46, 175-188

- Magand C., Ducharne A., Le Moine N. and Brigode P. (2015) Parameter transferability under changing climate: case study with a land surface model in the Durance watershed, France, Hydrolog. Sci. J., 60, 1408-1423

————————————————

REVIEWER

3) Throughout the paper, the term evapotranspiration (ET) is used, whereas the more general term 'evaporation' or 'total evaporation' may be more clear. I would like to refer to Savenije (2004) for some arguments to not use the term evapotranspiration as well. Briefly, transpiration is a rather different process compared to, for example, interception evaporation. Especially with regard to deforestation, it is important to make

this distinction, as it is probably transpiration that decreases.

AUTHORS

In the land-surface model community, the term evapotranspiration is commonly used and defined as the sum of soil evaporation, interception evaporation, and vegetation transpiration (Wang and Dickinson, 2012). Using the term "evaporation" may not be enough precise for all readers from different communities and could lead to misunderstandings and misinterpretations. Thus, we acknowledge the reviewer for the suggestions related to Savenije (2004) but would prefer using the evapotranspiration in our paper.

References cited:

- Wang, K. and Dickinson, R. E. A review of global terrestrial evapotranspiration: Observation, modeling, climatology, and climatic variability, Rev. Geophys., 2012, 50, RG200

—————————————

REVIEWER

Detailed comments: P4.L30. What do you mean with "business-as-usual"? The current situation? P12.L1. Idem

AUTHORS

The term "Business-as-usual" is commonly used for a reference scenario which is defined as a continuation of the current trend, here in term of deforestation. We added the definition in section 2.3. of the new manuscript.

—————————————

REVIEWER

P23.Fig3. Define the abbreviation of ASO (now only later in the text)

AUTHORS

ASO is already defined, and at its first use, at line 28 of page 6 in section 3.1.1. in the first version of the manuscript (now, line 4 of page 9 in section 3.1.1 of the new manuscript).

————————————————

REVIEWER

P33.Fig.14. Why is the third model not shown?

AUTHORS

INLAND-DGVM does not include a river routing scheme and thus cannot simulate river discharge. Only ORCHIDEE and LPJmL-DGVM are able to simulate discharge. To clarify this point, we added the information:

- in the text at line 24, page 12, in section 3.3.5 of the new manuscript.

- in the "Model setup" column of Table 1 of the new manuscript

- in the captions of Figures 12 and 13 of the new manuscript

Please also note the supplement to this comment:
http://www.hydrol-earth-syst-sci-discuss.net/hess-2016-430/hess-2016-430-AC1-supplement.pdf

**Supplement:**

[revised manuscript text omitted]

and the main processes controlling its dynamics are calculated from inputs of climate data (temperature, precipitation and cloud cover), atmospheric CO2, and soil texture. The main processes included in LPJmL-DGVM are the water balance, carbon balance, vegetation , establishment, phenology, mortality and fire disturbance.  The daily water balance of the soil is calculated by a simple bucket model, consisting of 5 soil layers of 20 cm, 30 cm, 50 cm, 1 m and 1 m depth, resulting in a cumulative depth of 3m. Water from precipitation that is not intercepted by vegetation enters the first soil layer depending on the amount of rainfall and the water saturation of the soil layer. The water that enters the first soil layer either evaporates, transpires or percolates to deeper soil layers. Evaporation from the canopy depends on the intercept water and the leaf area index of the vegetation. Evaporation from soil only occurs on bare soil and depends on the energy available for vaporization (potential evapotranspiration, PET). Plant transpiration is closely coupled to stomatal activity and photosynthesis and is calculated as a function of soil water supply and atmospheric demand (Sitch et al., 2003). All excess water above field capacity runs off as surface or subsurface runoff. The water is simulated to percolate from the first layer through the deeper soil layers based on a storage routine technique (Schaphoff et al., 2013) and is added to the runoff as baseflow component (Gerten et al., 2004b). The runoff is routed through a gridded river network (Vörösmarty et al., 2000), with a constant flow velocity of $1\,\mathrm{m\,s^{-1}}$ (Rost et al., 2008). Human processes like irrigation extraction and the operation of large reservoirs is explicitly accounted for (Rost et al., 2008; Biemans et al., 2011).  The carbon balance includes a detailed simulation of photosynthesis (based on Farquhar et al. (1980) and Collatz et al. (1992)), autotrophic and heterotrophic respiration, allocation of carbon to the plant compartments, establishment, mortality and phenology (Sitch et al., 2003). These processes are in LPJmL-DGVM calculated for  nine plant functional types (PFTs) representing natural vegetation for each grid cell. each PFT representing an assortment of species classified as being functionally similar. In this study for the Amazon basin, LPJmL-DGVM primarily simulates three of these plant functional types, representing tropical evergreen and deciduous forests and C4 grasses. LPJmL-DGVM also includes crop growth and harvest of so-called crop functional types on managed land as well as managed grassland (Bondeau et al., 2007). LPJmL-DGVM has been prove to reproduce observed patterns of biomass production, the global water balance, river discharge, tropical vegetation dynamics and fire (Cramer et al., 2001; Sitch et al., 2003; Wagner et al., 2003; Gerten et al., 2004a, 2008; Rost et al., 2008; Biemans et al., 2009; Poulter et al., 2009; Fader et al., 2010; Thonicke et al., 2010). It has been shown that the observed patterns in water fluxes (including soil moisture, evapotranspiration and runoff) are comparable to stand-alone global hydrological models (Wagner et al., 2003; Gerten et al., 2004a; Gordon et al., 2004; Gerten et al., 2008; Biemans et al., 2009; Haddeland et al., 2011). Several studies on Amazonia have been conducted showing the effect of climate change on NPP (Poulter et al., 2009), on carbon stocks (Gumpenberger et al., 2010), on the risk for forest dieback (Rammig et al., 2010) and also on patterns of inundation duration and inundated area (Langerwisch et al., 2013).

**INLAND-DGVM (INtegrated model of LAND surface processes)**

~~INLAND (Foley et al., 1996; Kucharik et al., 2000) is the land surface module of the Brazilian Earth System Model (BESM), and represents virtually all relevant aspects of the land surface to the climate system. BESM is a world-class global coupled model of the climate system currently being developed within Brazil's Climate Change Program that includes modules of atmospheric and ocean general circulation, the terrestrial and marine biosphere, cryosphere, carbon cycles, and aerosols. INLAND simulates 12 different PFTs competing for available resources~~

within the grid cell and the relative success of each PFT determines its fractional coverage. The model allows trees and herbaceous plants or grasses to experience different light and water availability: while trees in the upper canopy have priority in capturing available light (thus shading the shrubs and grasses in the lower part of the canopy), the herbaceous plants are able to capture soil water first when it infiltrates the ground (Foley et al., 1996). INLAND uses the mechanistic treatment of canopy photosynthesis proposed by Farquhar et al. (1980) and the semi-mechanistic Ball-Berry approach to estimate stomatal conductance (Ball et al., 1987; Collatz et al., 1992), computing gross photosynthesis, maintenance respiration and growth respiration to yield the annual carbon balance for each PFT, and the vegetation dynamics module simulates biomass changes for each PFT on a yearly time step. The model uses specific soil water-stress functions to down-regulate the gross primary productivity of vegetation as soils dry.

INLAND-DGVM is premised to be single, physically consistent model that solves the energy, water, carbon, and momentum balance of the soil-vegetation-atmosphere system and can be directly incorporated within Atmospheric Climate models. Based on the LSX package of Thompson and Pollard (1995), it represents canopy and soil physics processes by explicitly diagnosing the temperature of the vegetation in two canopy layers (e.g. trees versus shrubs and grasses) and of its soil layers, as well as air temperature and specific humidity within canopy air spaces, driven by the radiation balance of the vegetation and the ground, and the diffusive and turbulent fluxes of sensible heat and water vapor. In order to resolve the diurnal cycle, the model solves the canopy physics at its shortest time step (depending on the user choice, usually $30 - 60$ min). The total amount of evapotranspiration is treated as the sum of three water vapor fluxes: evaporation from the soil, evaporation of water intercepted by the vegetation and canopy transpiration.

The model state description includes 6 soil layers with varying thicknesses (to simulate the diurnal and seasonal variations of heat and moisture in the total soil depth) that are parameterized with biome-specific root biomass distributions of Jackson et al. (1996). This permits a different root length density for each layer in the profile.

The dynamics of soil volumetric water content are simulated for each layer. Soil moisture is based on Richards' flow equation, where the soil moisture change in time and space is a function of soil hydraulic conductivity, soil water retention curve, plant water uptake, and upper and lower boundary conditions. The water budget is controlled by the rate of infiltration (Green and Ampt, 1911), evaporation of water from the soil surface, the transpiration stream originating from plants, and redistribution of water in the profile. The modeling of water flow in unsaturated soils requires the description of water uptake by plant roots. Water uptake by roots is represented by a sink term in the macroscopic Richards equation and only considers stress due to dry conditions through a simple heuristic approach that represents the influence of soil water stress on gross photosynthesis rates (Foley et al., 1996). The drainage from the bottom soil layer is modeled assuming gravity drainage and neglects interactions with groundwater aquifers. Foley et al. (1996); Kucharik et al. (2000) give additional descriptions of the IBIS model land surface physics, which is essentially transferred unaltered to INLAND-DGVM.

**ORCHIDEE (ORganising Carbon and Hydrology In Dynamic EcosystEms)**

ORCHIDEE (Krinner et al., 2005) is the land component of the IPSL (Institut Pierre Simon Laplace) coupled climate model. It simulates the energy and water fluxes between the soil, the vegetation, and the atmosphere through the SECHIBA (Schématisation des Echanges Hydriques à l'Interface entre la Biosphère et l'Atmosphère, Ducoudré et al., 1993; de Rosnay and Polcher, 1998) module, while  andthe $CO_2$ fluxes and ecosystem carbon cycling throughare described by the STOMATE (Saclay Toulouse Orsay Model for the Analysis of Terrestrial Ecosystems, Viovy, 1996) module. When coupled with SECHIBA, STOMATE links the fast hydrological and biophysical processes with the carbon dynamics. STOMATE also contains a dynamic vegetation model, but this module was not activated for this study. In each grid cell, up to 12 plant functional types (PFTs) can be represented simultaneously, in addition to bare soil. LAI dynamics (from carbohydrate allocation) is simulated by STOMATE

which models the allocation of assimilates, autotrophic respiration components, foliar development, mortality and litter and soil organic matter decomposition. A factor representing drought stress (McMurtrie et al., 1990) linearly computes the rate of ribulose bisphosphate (RuBP) regeneration and the carboxylation rate.

The drought stress and the leaf age of the vegetation directly influence the photosynthetic capacity (Farquhar et al., 1980; Collatz et al., 1992; Verbeeck et al., 2011; de Weirdt et al., 2012), and  the stomatal conductance (Ball et al., 1987), which  controls the transpiration and is a function of two profiles: a fixed root density profile for each PFT, and the soil moisture profile (de Rosnay and Polcher, 1998). Canopy interception is proportional to LAI and the corresponding evaporation proceeds at potential rate, like the soil evaporation. In the latter case, however, soil moisture can become limiting if the upward diffusion to the top soil layer cannot supply enough water to sustain the required potential rate.

Soil moisture redistribution is described by a multi-layer scheme to solve the Richards equation for vertical unsaturated flow under the effect of root uptake (de Rosnay et al., 2002; Campoy et al., 2013). The hydraulic conductivity and diffusivity depend on soil moisture following the Van Genuchten (1980) model; the required parameters are taken from (Carsel and Parrish, 1988), and depend on the dominant soil texture in each grid-cell, based on the 1° × 1° texture map by Zobler (1986). The 2-m soil column is divided into 11 layers, with thickness increasing geometrically with depthwhile the saturated hydraulic conductivity exponentially decreases with depth, to account for increased compaction and reduced bioturbation (Beven and Kirkby, 1979). The precipitation rate and the soil hydraulic conductivity govern the partitioning between  surface runoff  and soil infiltration, which involves a time splitting procedure inspired from Green and Ampt (1911) to describe the propagation of the wetting front. The second contribution to total runoff is  gravitational drainage at the bottom of the soil.

The routing module (Polcher, 2003; Ngo-Duc et al., 2005; Guimberteau et al., 2012) calculates the daily discharge in each grid-cell and to the ocean. Streamflow routing relies on a series of linear reservoirs along the drainage network, derived from a 0.5° resolution data set (Vörösmarty et al., 2000). The routing scheme also includes a floodplain/swamp parameterization (d'Orgeval et al., 2008), recently improved by Guimberteau et al. (2012) for the Amazon basin, by introducing a new floodplain/swamp map. The simulation of the hydrology by the model ORCHIDEE has been widely tested over the Amazon basin and its catchments (Guimberteau et al., 2012; Getirana et al., 2014; Guimberteau et al., 2014).

| Location | Station Name | Abbreviation | River Name | Abbreviation | Latitude | Longitude | Area (km$^2$) |
|---|---|---|---|---|---|---|---|
| MAIN | Óbidos | OBI | Amazonas | AMAZ | -1.95 | -55.30 | 4,680,000 |
| SOUTH | Fazenda Vista Alegre | FVA | Madeira | MAD | -4.68 | -60.03 | 1,293,600 |
| | Guajará-Mirim | GMIR | Mamoré | MAM | -10.99 | -65.55 | 532,800 |
| | Itaituba | ITA | Tapajós | TAP | -4.24 | -56.00 | 461,100 |
| | Altamira | ALT | Xingu | XIN | -3.38 | -52.14 | 469,100 |
| WEST | Tamshiyacu | TAM | Upper Solimões | UPSO | -4.00 | -73.16 | 726,400 |
| | Lábrea | LAB | Purus | PUR | -7.25 | -64.80 | 230,000 |
| | Gavião | GAV | Juruá | JUR | -4.84 | -66.85 | 170,400 |
| NORTH | Caracaraí | CARA | Branco | BRA | +1.83 | -61.08 | 130,600 |

**Table S1.** List of the gauging stations for the studied catchments. Sources: SO HYBAM (Observation Service of the Geodynamical, hydrological and biogeochemical control of erosion/alteration and material transport in the Amazon, Orinoco and Congo basins, Cochonneau et al., 2006).

| | Basin | Model | Relative bias (%) | | Correlation coefficient | | NRMSE (%) | |
|---|---|---|---|---|---|---|---|---|
| | | | Q | ET | Q | ET | Q | ET |
| MAIN | AMAZ | INLAND-DGVM | -22.4 | -1.8 | - | 0.60 | - | 14.1 |
| | | LPJmL-DGVM | -21.9 | +1.9 | 0.77 | 0.55 | 36.6 | 25.0 |
| | | ORCHIDEE | -5.9 | -4.6 | 0.91 | 0.58 | 14.1 | 17.2 |
| SOUTH | MAD | INLAND-DGVM | -28.3 | +0.2 | - | 0.89 | - | 13.6 |
| | | LPJmL-DGVM | -2.2 | -9.5 | 0.89 | 0.83 | 33.5 | 28.9 |
| | | ORCHIDEE | -5.5 | -1.7 | 0.99 | 0.88 | 20.2 | 15.2 |
| | MAM | INLAND-DGVM | -60.9 | +0.92 | - | 0.99 | - | 15.0 |
| | | LPJmL-DGVM | +14.0 | -14.8 | 0.73 | 0.91 | 47.4 | 30.6 |
| | | ORCHIDEE | -22.2 | -3.0 | 0.91 | 0.98 | 43.4 | 18.0 |
| | TAP | INLAND-DGVM | +10.5 | -3.0 | - | 0.02 | - | 13.4 |
| | | LPJmL-DGVM | +25.1 | -6.8 | 0.90 | 0.45 | 53.9 | 34.0 |
| | | ORCHIDEE | +16.6 | -3.3 | 0.96 | 0.11 | 47.5 | 11.5 |
| | XIN | INLAND-DGVM | +47 | -1.9 | - | 0.17 | - | 14.9 |
| | | LPJmL-DGVM | +59.1 | -5.9 | 0.83 | -0.01 | 66.6 | 34.5 |
| | | ORCHIDEE | +34.1 | -4.4 | 0.94 | 0.31 | 46.1 | 12.6 |
| WEST | UPSO | INLAND-DGVM | -57.4 | +2.2 | - | 0.32 | - | 22.9 |
| | | LPJmL-DGVM | -45.2 | -0.9 | 0.93 | 0.87 | 86.0 | 18.5 |
| | | ORCHIDEE | -17.2 | -10.0 | 0.96 | 0.31 | 23.9 | 25.5 |
| | PUR | INLAND-DGVM | +9.3 | +2.6 | - | 0.83 | - | 9.8 |
| | | LPJmL-DGVM | +18.6 | +1.7 | 0.86 | 0.27 | 39.3 | 24.0 |
| | | ORCHIDEE | +15.8 | -0.9 | 0.96 | 0.79 | 31.6 | 10.1 |
| | JUR | INLAND-DGVM | +9.3 | -0.05 | - | 0.86 | - | 9.3 |
| | | LPJmL-DGVM | +10.2 | +7.3 | 0.89 | 0.02 | 29.7 | 17.0 |
| | | ORCHIDEE | +39.4 | -4.1 | 0.96 | 0.82 | 40.1 | 10.4 |
| NORTH | BRA | INLAND-DGVM | +47.1 | +17.1 | - | 0.74 | - | 21.0 |
| | | LPJmL-DGVM | +53.3 | +12.5 | 0.99 | 0.06 | 51.3 | 33.1 |
| | | ORCHIDEE | +69.3 | +10.9 | 0.96 | 0.61 | 58.8 | 15.0 |

**Table S2.** Bias (%), correlation and NRMSE (Normalized Root Mean Square Error) (%) against the observations, of discharge and ET, for each catchment, for HIST period. Observed discharge comes from SO HYBAM and ET is estimated by the product of Jung et al. (2010).

[Figure]

**Figure S1.** Deforested area (%) in each 25 x 25 km$^2$ for the LCC scenarios LODEF (**a** and **d**), HIDEF (**b** and **e**) and EXDEF (**c** and **f**).

[Figure]

**Figure S2.** Decrease of forest fraction for the three LCC scenarios (for the two time periods) compared with the NODEF scenario in 2009 over the Amazon basin. Grey colour indicates no change of forest fraction.

[Figure]

**Figure S3.** Seasonal change in ET (mm month$^{-1}$) due to deforestation combined with climate change (EXDEF) simulated by the three LSMs over the Amazon basin and its catchments, averaged over the two future periods. For a given LSM and period, the shaded area defines the envelope enclosing the range with plausible climate futures.

[Figure]

**Figure S24.** For each GCM-forcing, monthly mean seasonalities of the water budget components (including the ET components) (mm d$^{-1}$) from the three LSMs (rows) and for each NODEF and LCC scenarios (columns) over **(a)** the Madeira and **(b)** the Tapajós catchments. The variables of the water budget are: precipitation (P), runoff (R) and evapotranspiration (ET). The variables of the ET components are: transpiration (Tr), soil evaporation (Esoil) and evaporation of canopy interception (Ecanop).

---

## Author Comment (AC2) · 28 Nov 2016

In addition to our answers to the reviewers, we re-wrote a new version of the manuscript which is available in Supplement. In this new version, the additional text is in blue and the deleted one in red.

REVIEWER

Scientific quality:

1) More details on the GCM bias correction and statistical downscaling methods should

be provided. Since the authors are evidently using others' results the methodology descriptions do not need to be exhaustive but the methods used in the referenced papers should be identified so readers do not have to look up the previous work.

AUTHORS

You are right. An additional sub-section (2.2 Climate change scenarios) in the section "Materials and methods" describes now in more details the GCM bias correction and statistical downscaling.

————————-

REVIEWER

2) The LSMs are not adequately or consistently described in the manuscript or supplemental material. It is not clear how the important hydrologic processes are represented, or how they are parameterized, in the three LSMs. Details regarding the spatial and temporal resolutions of the models should be presented in Table 2, along with much more detail regarding how particular hydrologic processes (evaporation, transpiration, unsaturated flow, groundwater flow, overland flow, river routing) are simulated in the models. The description of the models in the supplementary material is qualitative rather than quantitative and focuses most strongly on vegetative processes.

AUTHORS

For each model, we revised the section "Models" in the Supplementary Material and we focused more on the description of the hydrology modeling rather than the vegetation. Moreover, we added the informations of time and spatial resolutions for each model in Table 1 in the new manuscript.

————————-

REVIEWER

3) No evidence is given that the LSMs are able to adequately simulate historic ET and

river fluxes in the study region. The supplemental materials broadly states that two of the three models (LPJmL-DGVM and ORCHIDEE) have been widely tested, but a comparative summary of the three models predictions during the historic period should be presented. Historical rivers flows are included in Figure 14 but no attempt is made to attribute errors in predictions or discuss the relative magnitude of errors among LSMs. Section 3.3.3 discusses differences in LSM predictions in a qualitative way but without knowing specifics of how the processes are simulated in the different models it is difficult to generalize the results beyond this modeling exercise. Since the goal of this manuscript is to understand hydrologic change it is important to show that the hydrologic models make credible predictions of ET, soil moisture, groundwater levels, river flow during the historical time period, and to understand how specific differences in hydrologic representation among the models lead to differences in predictions.

AUTHORS

The three models used in our study are not hydrological models as mentioned in your last sentence. As discussed in point [3] of the response to the Reviewer 3, these LSMs were not calibrated for their hydrological and river routing parameters. In this paper, LSMs are used to evaluate the hydrological response of the vegetation to the climate change and land cover change which cannot be represented by most classical hydrological models. In the new Table S2 in Supplementary Material of the new manuscript, we give the results of comparison (Relative bias, Correlation and NRMSE) between ET and runoff simulated by the three models in present time and the observations. For ET comparison, we used the machine-learning FLUXNET product (Jung et al., 2010) itself uncertain for the Amazon basin because given the small number of flux tower measurements available. The evaluation of historical ET and runoff simulated by the different models over the Amazon basin can be also found in the literature:

- for ORCHIDEE: Guimberteau et al. (2012, 2014)

- for LPJmL-DGVM: Langerwisch et al. (2013)

- for INLAND-DGVM: Dias et al. (2015), Lyra et al. (2016)

We introduced Table S2 and cite these references in section 2.1. of the new manuscript.

References cited:

- Dias L. C. P., Macedo M., Marcia N., Costa M. H., Coe M. T. and Neill C. (2015) Effects of land cover change on evapotranspiration and streamflow of small catchments in the Upper Xingu River Basin, Central Brazil, J. Hydrol.: Reg. Stud., 4, 108-122

- Guimberteau M., Drapeau G., Ronchail J., Sultan B., Polcher J., Martinez J. M., Prigent C., Guyot J. L., Cochonneau G., Espinoza J. C., Filizola N., Fraizy P., Lavado W., De Oliveira E., Pombosa R., Noriega L. and Vauchel, P. (2012) Discharge simulation in the sub-basins of the Amazon using ORCHIDEE forced by new datasets, Hydrol. Earth Syst. Sc., 16, 911-935

- Guimberteau M., Ducharne A., Ciais P., Boisier J.-P., Peng S., De Weirdt M. and Verbeeck H. (2014) Testing conceptual and physically based soil hydrology schemes against observations for the Amazon Basin, Geosci. Model Dev., 7, 1115-1136

- Jung, M.; Reichstein, M.; Ciais, P.; Seneviratne, S.; Sheffield, J.; Goulden, M.; Bonan, G.; Cescatti, A.; Chen, J.; De Jeu, R.; Johannes Dolman, A.; Eugster, W.; Gerten, D.; Gianelle, D.; Gobron, N.; Heinke, J.; Kimball, J.; Law, B. E.; Montagnani, L.; Mu, Q.; Mueller, B.; Oleson, K.; Papale, D.; Richardson, A. D.; Roupsard, O.; Running, S.; Tomelleri, E.; Viovy, N.; Weber, U.; Williams, C.; Wood, E.; Zaehle, S. & Zhang, K. Recent decline in the global land evapotranspiration trend due to limited moisture supply, Nature, 2010, 467, 951-954

- Langerwisch F., Rost S., Gerten D., Poulter B., Rammig A. and Cramer W. (2013) Potential effects of climate change on inundation patterns in the Amazon Basin, Hydrol. Earth Syst. Sc., 17, 2247-2262

- Lyra A. d. A., Chou S. C. and Sampaio G. d. O. (2016) Sensitivity of the Amazon

biome to high resolution climate change projections, Acta Amazon., 46, 175-188

————————-

REVIEWER

4) The description of the development of the deforestation scenarios is somewhat confusing. I am not a land use change modeler but it is not clear to me how the results from the participatory process in certain parts of the study region were extrapolated and incorporated into the LuccME and CLUE models, or why two different land use change models were required. The manuscript states (e.g. line 21 p 5 and line 34 p 5) that maps were used to "calibrate and validate the deforestation model", and scenarios were "translated into model parameters" but no details are given on the methods, parameter values or prediction accuracies. I am left wondering how good the land use change models are, and how much the precise spatial distribution of deforestation shown in figure 7 matters, versus a more generic uniform decrease in forest area across the domain. If the actual locations of increased deforestation make a difference to predictions of future hydrologic change this would be interesting.

AUTHORS

We agree with the reviewer. We rewrote section 2.3. to clarify the following points:

- About the participatory process, there was no extrapolation. Two stakeholder workshops were held for discussing the whole Brazilian Amazon future, along four axes: natural resources, social development, economic activities and institutional context. The results are multi-dimensional and rich qualitative storylines. To feed the spatial model of land use, only some selected elements of the storylines were used - mainly concerning to the natural resources theme: (a) deforestation rates; (b) secondary vegetation dynamics; (c) roads and protected areas network; (d) law enforcement. The quantification process for the Brazilian Amazon is described in Aguiar et al. (2016). For the Bolivian Amazon, expert-driven premises about these same selected elements

were adopted - respecting however the Bolivian socioeconomic and political specificities, as explained in Tejada et al. (2015). There were no resources in the project to repeat the participatory process, and it was actually not mandatory to parametrize the model.

- About LuccME and CLUE, we apologize for not explaining it correctly in the earlier version of the manuscript. LuccME is an open source modeling framework that implements a version of the CLUE model. We made it more clear in the manuscript.

- The reason for having regionalized the spatial model (Brazil, Bolivia and the other countries) is now explained in the text. The Amazon drainage basin covers an area of about 7,500,000 km2 (2,900,000 sq mi), or roughly 40 percent of the South American continent. It is located in the countries of Bolivia, Brazil, Colombia, Ecuador, Peru and Venezuela. Each country in the basin has its own socioeconomic and institutional context, and therefore specific aspects to be taken into consideration when building scenarios. To avoid oversimplifications, our choice was generating updated scenarios only in Brazil and Bolivia, the most important deforestation hotspots in the basin. The Brazilian portion of the basin covers approximately 50% of the area, being also where most of the deforestation hotspots have been located in the previous decades. Bolivia has also been facing an intensive deforestation process for agricultural expansion around the Santa Cruz area. For the other countries, existing spatial projections were used.

- We added some additional methodological information in the manuscript. Aguiar et al., (2016) and Tejada et al. (2015) provide further detail about the quantification process, including calibration and validation steps, considering observed spatial patterns. One important thing is that scenarios are not predictions (Raskin et al., 2005), "They are about envisioning future pathways and accounting for critical uncertainties". Therefore, our hydrological analysis results are valid and sound considering the selected land use scenarios and their underlying premises.

————————-

REVIEWER

The authors present a wide range of results showing how future projections differ based on GCM, LSM and LCC scenario. However it is unclear at the end of my reading of the manuscript what are the dominant drivers of these differences, or what new insights or actionable information has been generated from this study. It would be useful if the authors could synthesize their results to quantitatively apportion uncertainty for various hydrologic predictions among the three sources (GCM, LSM, LCC).

AUTHORS

Thank you for this helpful remark. With the ANOVA framework, we decomposed in the revised manuscript the overall uncertainty in future runoff/ET projections into the fraction of uncertainty that is related/explained by GCMs (climate change uncertainty in our framework), LSMs, LCC scenarios, and by the interactions between these factors. The new Figure 14 gives the different contributions of these factors to total uncertainty in ET and runoff changes for the Amazon basin and eight catchments. We rewrote the section 4.3. and added information in the conclusion to highlight the dominant uncertainty in ET and runoff projections among the three sources.

————————-

REVIEWER

It might also be interesting to weight the ensemble of future projections based on historical reliability of the GCMs and LSMs and possibly the convergence of their future predictions (see e.g. reliability ensemble averaging (e.g. Giorgi and Mearns, Geophysical Research Letters, 2003; Asefa and Adams, Regional Climate Change 2013).

AUTHORS

Thank you for this remark but the choice to weight the ensemble of future projections

is very controversial (Knutti et al., 2010). In most studies the weighted averages and model spread are similar to those of the unweighted ensemble due to the absence of correlation between the observations used to weight the models and the models' future projections (Knutti et al., 2010). Moreover, in our study, we have only three elements for each source of uncertainties which is not sufficient to weight the ensemble. Thus, we decided not to weight the ensemble of future projections for our study.

References cited:

- Knutti, R. The end of model democracy?, Clim. Change, 2010, 102, 395-404

- Knutti, R.; Furrer, R.; Tebaldi, C.; Cermak, J. & Meehl, G. A. Challenges in Combining Projections from Multiple Climate Models, J. Climate, 2010, 23, 2739-2758

————————-

REVIEWER

Presentation quality:

[. . .] many of the labels and legends are difficult to read (e.g. Fig 1a, Fig 4 , Fig 8, Fig 13 ). In addition the figures are quite numerous. It would be helpful if the figures could be reduced and their content synthesized to more succinctly present the study's major findings and conclusions.

AUTHORS

- We improved the labels and legends in the figures in the new manuscript:

* Fig 1a: the axis labels and ticks are made clearer.

* Figs 4 and 7: as suggested by the reviewer 3, we use now one set of legends instead of four with a larger font

* Fig 11: the lines of ET, Runoff and Transpiration are thicker. We changed the color of Transpiration line by a darker green.

**HESSD**

- We put the Figures 7 and 12 of the first version of the manuscript in Supplementary Material (now Figs S2 and S3).

Please also note the supplement to this comment:
http://www.hydrol-earth-syst-sci-discuss.net/hess-2016-430/hess-2016-430-AC2-supplement.pdf

―――――――――――――――――

---

## Author Comment (AC3) · 28 Nov 2016

In addition to our answers to the reviewers, we re-wrote a new version of the manuscript which is available in Supplement. In this new version, the additional text is in blue and the deleted one in red.

REVIEWER

Major Comments:

[1] Use of the new LCC scenarios is novel in your study. Therefore, it is important to

provide some literature review indicating how different your results are compared to the previous studies. You discuss Siqueira Júnior et al. (2015) in Page 11 Line 32. I suggest you provide this type of examples more often so that the manuscript is well set within the context of the existing literature.

AUTHORS

We agree with the reviewer. In the introduction, we included a more comprehensive explanation about the difference from the previous studies concerning land use scenario modeling. We also explain the need of updated scenarios given the changes in the deforestation dynamics during the last decade.

Concerning the response of the hydrology to Amazon deforestation, there have been several studies addressing the Amazon deforestation effects with models since the late 1980s. Most of these papers describe climate impacts, notably on rainfall (Lejeune et al., 2016; Spracklen and Garcia-Carreras, 2015) based on GCM (coupled) simulations and did not really focus on surface hydrology. In offline mode (thus with an inconsistent treatment of atmospheric feedbacks between LSM water fluxes and CCM water fluxes), the studies published in the literature focused on deforestation effect only for the historical periods.

We found one very recent study (Lamparter et al., 2016) analyzing the impact of Amazon deforestation on the hydrology in offline models using four different scenarios (trend, sustainable, legal and illegal) provided by the LANDSHIFT model during the Carbiocial project. The trend scenario showed an increase of low flows in an upper catchment of the Tapajos, in agreement with our findings. We added this reference in the discussion, in section 4.2.

References cited:

- Lamparter, G.; Nobrega, R. L. B.; Kovacs, K.; Amorim, R. S. & Gerold, G. Modelling hydrological impacts of agricultural expansion in two macro-catchments in Southern

Amazonia, Brazil, Reg. Environ. Change, 2016

- Lejeune, Q.; Davin, E. L.; Guillod, B. P. & Seneviratne, S. I. Influence of Amazonian deforestation on the future evolution of regional surface fluxes, circulation, surface temperature and precipitation, Clim. Dyn., 2014, 44, 2769-2786

- Spracklen, D. and Garcia-Carreras, L. The impact of Amazonian deforestation on Amazon basin rainfall, Geophys. Res. Lett., 2015, 42, 9546-9552

————————————

REVIEWER

[2] Include a discussion on why you selected the three models. I have seen that you have provided details about individual models in the supplementary materials, however, a discussion is warranted on why these three particular models were selected. How are the models different from each other and why does that difference matter? Discuss why you think that multi-model approach is better than selecting the single best model (based on historical simulation performance).

AUTHORS

The three models used in our study represent the state of the art in global land surface modeling. ORCHIDEE and LPJmL-DGVM are used e.g. for ISIMIP project (The Inter-Sectoral Impact Model Intercomparison Project, https://www.isimip.org/) that aimed at contributing to a quantitative and cross-sectoral synthesis of the differential impacts of climate change, including the associated uncertainties. INLAND-DGVM has been widely tested over South American biomes to represent the biosphere-atmosphere interactions. Thus, the three LSMs are representative of the diversity of approaches to describe the functioning of the coupled system vegetation-hydrology. Moreover, two out of three models integrate different river routing schemes and are thus able to simulate the change of river discharge with climate change and in interaction with the land cover change. We added this information in section 2.1 of the new version of the manuscript.

We would like to mention also that the best model in present time does not give necessary trustable results in a climatic change perspective, as reported by e.g. Habets et al. (2013). In addition, dominant hydrological processes (and sources of uncertainty from each process) can change between present and future. With the CLSM model, Magand et al. (2015) found that the parameters controlling soil moisture had more influence in the future than in present time, for instance. Under climatic change conditions, they found that the dominant process was no longer related to snow but rather to evapotranspiration model equations. Thus, that is why we did not trust any specific model, and we preferred to adopt a multi-model approach in our paper, recommended by Knutti (2010).

References cited:

- Habets F., Boé J., Déqué M., Ducharne A., Gascoin S., Hachour A., Martin E., Pagé C., Sauquet E., Terray L., Thiéry D., Oudin L. and Viennot P. (2013) Impact of climate change on the hydrogeology of two basins in northern France, Clim. Change, 121, 771-785

- Knutti, R. (2010) The end of model democracy?, Climatic Change, 102:395. doi:10.1007/s10584-010-9800-2

- Magand C., Ducharne A., Le Moine N. and Brigode P. (2015) Parameter transferability under changing climate: case study with a land surface model in the Durance watershed, France, Hydrolog. Sci. J., 60, 1408-1423

————————

REVIEWER

[3] Are the LSMs calibrated? It is really difficult to trust a model if it is not calibrated and evaluated. As you know, model outcomes are subject to vary (often significantly) if the parameter values are changed, given that the model structure is fixed. So, model calibration/evaluation is crucial for any model simulation-based studies.

AUTHORS

The LSMs were not calibrated by adjusting hydrological or water routing parameters. Calibration of LSMs with a routing scheme and rather complex soil-plant water transfer models is a difficult task because of the much higher number of parameters than in hydrological (catchment hydrology) models. Moreover, calibration is site and time specific and does not ensure a good behavior of a LSM in very large catchments, the domain of application of the LSMs. We added the non-calibration information in section 2.1 of the new version of the manuscript.

There are several sources of uncertainty for hydrological models simulations of hydrological change under climate change, in particular due to the calibration. Brigode et al. (2013) found that the hydrological model robustness was the major source of variability in streamflow projections (for 89 catchments in northern and central France) in future climate conditions, leading to difficulties to calibrate hydrological models for climate change studies (i.e. when the climatic space between calibration and validation periods is different). This result corroborates with the findings of Vaze et al. (2010), Merz et al. (2011) and Coron et al. (2012) in other catchments.

Moreover, as we said in the previous point, a "realistic model" in present time does not give necessary realistic results in a climatic change perspective.

References cited:

- Brigode P., Oudin L. and Perrin C. (2013) Hydrological model parameter instability: A source of additional uncertainty in estimating the hydrological impacts of climate change?, J. Hydrol., 476, 410-425.

- Coron, L., Andréassian, V., Perrin, C., Lerat, J., Vaze, J., Bourqui, M., Hendrickx, F., 2012. Crash testing hydrological models in contrasted climate conditions: an experiment on 216 Australian catchments. Water Resour. Res.. http://dx.doi.org/10.1029/2011WR011721.
- Merz, R., Parajka, J., Blöschl, G., 2011. Time stability of catchment model parameters: implications for climate impact analyses. Water Resour. Res. 47, W02531. http://dx.doi.org/10.1029/2010WR009505.

- Vaze, J., Post, D.A., Chiew, F.H.S., Perraud, J.M., Viney, N.R., Teng, J., 2010. Climate non-stationarity – validity of calibrated rainfall-runoff models for use in climate change studies. J. Hydrol. 394 (3–4), 447–457, doi : 16/j.jhydrol.2010.09.018.

————————————

REVIEWER

Having said that, I understand that calibrating three LSMs might be difficult. However, you should at least show how consistent the model simulations are. For this study, it would be essential to compare the historical simulations of discharge and ET from all three models against the observed data.

AUTHORS

In the new Table S2 in Supplementary Material of the new manuscript, we give the results of comparison (Relative bias, Correlation and NRMSE) between ET and runoff simulated by the three models in present time and the observations. For ET comparison, we used the machine-learning FLUXNET product (Jung et al., 2010) itself uncertain for the Amazon basin because given the small number of flux tower measurements available.

The evaluation of historical ET and runoff simulated by the different models over the Amazon basin can be also found in the literature:

- for ORCHIDEE: Guimberteau et al. (2012, 2014)

- for LPJmL-DGVM: Langerwisch et al. (2013)

- for INLAND-DGVM: Dias et al. (2015), Lyra et al. (2016)

[Figure]

We introduced Table S2 and cite these references in section 2.1. of the new manuscript.

- Dias L. C. P., Macedo M., Marcia N., Costa M. H., Coe M. T. and Neill C. (2015) Effects of land cover change on evapotranspiration and streamflow of small catchments in the Upper Xingu River Basin, Central Brazil, J. Hydrol.: Reg. Stud., 4, 108-122

- Guimberteau M., Drapeau G., Ronchail J., Sultan B., Polcher J., Martinez J. M., Prigent C., Guyot J. L., Cochonneau G., Espinoza J. C., Filizola N., Fraizy P., Lavado W., De Oliveira E., Pombosa R., Noriega L. and Vauchel, P. (2012) Discharge simulation in the sub-basins of the Amazon using ORCHIDEE forced by new datasets, Hydrol. Earth Syst. Sc., 16, 911-935

- Guimberteau M., Ducharne A., Ciais P., Boisier J.-P., Peng S., De Weirdt M. and Verbeeck H. (2014) Testing conceptual and physically based soil hydrology schemes against observations for the Amazon Basin, Geosci. Model Dev., 7, 1115-1136

- Jung, M.; Reichstein, M.; Ciais, P.; Seneviratne, S.; Sheffield, J.; Goulden, M.; Bonan, G.; Cescatti, A.; Chen, J.; De Jeu, R.; Johannes Dolman, A.; Eugster, W.; Gerten, D.; Gianelle, D.; Gobron, N.; Heinke, J.; Kimball, J.; Law, B. E.; Montagnani, L.; Mu, Q.; Mueller, B.; Oleson, K.; Papale, D.; Richardson, A. D.; Roupsard, O.; Running, S.; Tomelleri, E.; Viovy, N.; Weber, U.; Williams, C.; Wood, E.; Zaehle, S. & Zhang, K. Recent decline in the global land evapotranspiration trend due to limited moisture supply} Nature, Nature Publishing Group, 2010, 467, 951-954

- Langerwisch F., Rost S., Gerten D., Poulter B., Rammig A. and Cramer W. (2013) Potential effects of climate change on inundation patterns in the Amazon Basin, Hydrol. Earth Syst. Sc., 17, 2247-2262

- Lyra A. d. A., Chou S. C. and Sampaio G. d. O. (2016) Sensitivity of the Amazon biome to high resolution climate change projections, Acta Amazon., 46, 175-188
* * *
REVIEWER

You show historical discharge in Fig. 14, however, one of the three models is missing.

AUTHORS

INLAND-DGVM does not include a river routing scheme and thus cannot simulate river discharge. Only ORCHIDEE and LPJmL-DGVM are able to simulate discharge. To clarify this point, we added the information:

- in the text at line 24, page 12, in section 3.3.5 of the new manuscript.

- in the "Model setup" column of Table 1 of the new manuscript

- in the captions of Figures 12 and 13 of the new manuscript

———————————

REVIEWER

Minor Comments:

[1] Page 2 Line 24: Please add citations.

AUTHORS

Added in the text.

———————————

REVIEWER

[2] Page 2 Line 27: Please add citations.

AUTHORS

Added in the text.

———————————

[Figure]

REVIEWER

[3] Page 3 Line 16: Please state what you have done more clearly. Whether you have used a 3D matrix or not is probably not that important in the introduction section. I suggest you avoid this type of technical detailing in the introduction section and focus on them more in the 'Materials and Methods' section.

AUTHORS

We modified the last sentence of the introduction section to state more clearly what we have done.

———————————

REVIEWER

[4] Page 6 Line 2: Any specific reason for selecting these two periods?

AUTHORS

We chose these two periods to have a focus on the middle and the end of the century which present two levels of deforestation very different in our scenarios.

———————————

REVIEWER

[5] Page 8 Line 18: I see a huge difference in between-LSM ET simulations. I wonder how much of this variability is attributed to improper model calibration. Any comments?

AUTHORS

LSMs are not calibrated and cannot be at such a large scale, especially for ET because of the high uncertainties in the ET products.

———————————

REVIEWER

[6] Page 31 Fig 12: How is the 'range' defined here?

AUTHORS

For a given LSM and each month, the range is defined as the minimum and the maximum values of simulated ET between the three climate change scenarios.

——————————

REVIEWER

[7] Page 33 Fig 14: Why are there only two models?

AUTHORS

INLAND-DGVM does not include a river routing scheme and thus cannot simulate river discharge. Only ORCHIDEE and LPJmL-DGVM are able to simulate discharge. To clarify this point, we added the information:

- in the text at line 24, page 12, in section 3.3.5 of the new manuscript.

- in the "Model setup" column of Table 1 of the new manuscript

- in the captions of Figures 12 and 13 of the new manuscript

——————————

REVIEWER

Figure Related Comments:

[1] Fig 1a: The axis labels and ticks are not very clear.

AUTHORS

The axis labels and ticks are made clearer.

——————————

REVIEWER

[2] Fig 8: The plot legends are too small. Maybe use one set of legends instead of four with a larger font?

AUTHORS

Yes, you are right. We use now one set of legends.

————————————

REVIEWER

[3] Fig 13: Maybe change the color for transpiration?

AUTHORS

Done in this figure. Moreover, the lines of ET, Runoff and Transpiration are thicker for better clarity.

Please also note the supplement to this comment:
http://www.hydrol-earth-syst-sci-discuss.net/hess-2016-430/hess-2016-430-AC3-supplement.pdf

**Supplement:**

[revised manuscript text omitted]

and the main processes controlling its dynamics are calculated from inputs of climate data (temperature, precipitation and cloud cover), atmospheric CO2, and soil texture. The main processes included in LPJmL-DGVM are the water balance, carbon balance, vegetation , establishment, phenology, mortality and fire disturbance.  The daily water balance of the soil is calculated by a simple bucket model, consisting of 5 soil layers of 20 cm, 30 cm, 50 cm, 1 m and 1 m depth, resulting in a cumulative depth of 3m. Water from precipitation that is not intercepted by vegetation enters the first soil layer depending on the amount of rainfall and the water saturation of the soil layer. The water that enters the first soil layer either evaporates, transpires or percolates to deeper soil layers. Evaporation from the canopy depends on the intercept water and the leaf area index of the vegetation. Evaporation from soil only occurs on bare soil and depends on the energy available for vaporization (potential evapotranspiration, PET). Plant transpiration is closely coupled to stomatal activity and photosynthesis and is calculated as a function of soil water supply and atmospheric demand (Sitch et al., 2003). All excess water above field capacity runs off as surface or subsurface runoff. The water is simulated to percolate from the first layer through the deeper soil layers based on a storage routine technique (Schaphoff et al., 2013) and is added to the runoff as baseflow component (Gerten et al., 2004b). The runoff is routed through a gridded river network (Vörösmarty et al., 2000), with a constant flow velocity of $1\,\mathrm{m\,s^{-1}}$ (Rost et al., 2008). Human processes like irrigation extraction and the operation of large reservoirs is explicitly accounted for (Rost et al., 2008; Biemans et al., 2011).  The carbon balance includes a detailed simulation of photosynthesis (based on Farquhar et al. (1980) and Collatz et al. (1992)), autotrophic and heterotrophic respiration, allocation of carbon to the plant compartments, establishment, mortality and phenology (Sitch et al., 2003). These processes are in LPJmL-DGVM calculated for  nine plant functional types (PFTs) representing natural vegetation for each grid cell. each PFT representing an assortment of species classified as being functionally similar. In this study for the Amazon basin, LPJmL-DGVM primarily simulates three of these plant functional types, representing tropical evergreen and deciduous forests and C4 grasses. LPJmL-DGVM also includes crop growth and harvest of so-called crop functional types on managed land as well as managed grassland (Bondeau et al., 2007). LPJmL-DGVM has been prove to reproduce observed patterns of biomass production, the global water balance, river discharge, tropical vegetation dynamics and fire (Cramer et al., 2001; Sitch et al., 2003; Wagner et al., 2003; Gerten et al., 2004a, 2008; Rost et al., 2008; Biemans et al., 2009; Poulter et al., 2009; Fader et al., 2010; Thonicke et al., 2010). It has been shown that the observed patterns in water fluxes (including soil moisture, evapotranspiration and runoff) are comparable to stand-alone global hydrological models (Wagner et al., 2003; Gerten et al., 2004a; Gordon et al., 2004; Gerten et al., 2008; Biemans et al., 2009; Haddeland et al., 2011). Several studies on Amazonia have been conducted showing the effect of climate change on NPP (Poulter et al., 2009), on carbon stocks (Gumpenberger et al., 2010), on the risk for forest dieback (Rammig et al., 2010) and also on patterns of inundation duration and inundated area (Langerwisch et al., 2013).

**INLAND-DGVM (INtegrated model of LAND surface processes)**

~~INLAND (Foley et al., 1996; Kucharik et al., 2000) is the land surface module of the Brazilian Earth System Model (BESM), and represents virtually all relevant aspects of the land surface to the climate system. BESM is a world-class global coupled model of the climate system currently being developed within Brazil's Climate Change Program that includes modules of atmospheric and ocean general circulation, the terrestrial and marine biosphere, cryosphere, carbon cycles, and aerosols. INLAND simulates 12 different PFTs competing for available resources~~

within the grid cell and the relative success of each PFT determines its fractional coverage. The model allows trees and herbaceous plants or grasses to experience different light and water availability: while trees in the upper canopy have priority in capturing available light (thus shading the shrubs and grasses in the lower part of the canopy), the herbaceous plants are able to capture soil water first when it infiltrates the ground (Foley et al., 1996). INLAND uses the mechanistic treatment of canopy photosynthesis proposed by Farquhar et al. (1980) and the semi-mechanistic Ball-Berry approach to estimate stomatal conductance (Ball et al., 1987; Collatz et al., 1992), computing gross photosynthesis, maintenance respiration and growth respiration to yield the annual carbon balance for each PFT, and the vegetation dynamics module simulates biomass changes for each PFT on a yearly time step. The model uses specific soil water-stress functions to down-regulate the gross primary productivity of vegetation as soils dry.

INLAND-DGVM is premised to be single, physically consistent model that solves the energy, water, carbon, and momentum balance of the soil-vegetation-atmosphere system and can be directly incorporated within Atmospheric Climate models. Based on the LSX package of Thompson and Pollard (1995), it represents canopy and soil physics processes by explicitly diagnosing the temperature of the vegetation in two canopy layers (e.g. trees versus shrubs and grasses) and of its soil layers, as well as air temperature and specific humidity within canopy air spaces, driven by the radiation balance of the vegetation and the ground, and the diffusive and turbulent fluxes of sensible heat and water vapor. In order to resolve the diurnal cycle, the model solves the canopy physics at its shortest time step (depending on the user choice, usually $30 - 60$ min). The total amount of evapotranspiration is treated as the sum of three water vapor fluxes: evaporation from the soil, evaporation of water intercepted by the vegetation and canopy transpiration.

The model state description includes 6 soil layers with varying thicknesses (to simulate the diurnal and seasonal variations of heat and moisture in the total soil depth) that are parameterized with biome-specific root biomass distributions of Jackson et al. (1996). This permits a different root length density for each layer in the profile.

The dynamics of soil volumetric water content are simulated for each layer. Soil moisture is based on Richards' flow equation, where the soil moisture change in time and space is a function of soil hydraulic conductivity, soil water retention curve, plant water uptake, and upper and lower boundary conditions. The water budget is controlled by the rate of infiltration (Green and Ampt, 1911), evaporation of water from the soil surface, the transpiration stream originating from plants, and redistribution of water in the profile. The modeling of water flow in unsaturated soils requires the description of water uptake by plant roots. Water uptake by roots is represented by a sink term in the macroscopic Richards equation and only considers stress due to dry conditions through a simple heuristic approach that represents the influence of soil water stress on gross photosynthesis rates (Foley et al., 1996). The drainage from the bottom soil layer is modeled assuming gravity drainage and neglects interactions with groundwater aquifers. Foley et al. (1996); Kucharik et al. (2000) give additional descriptions of the IBIS model land surface physics, which is essentially transferred unaltered to INLAND-DGVM.

**ORCHIDEE (ORganising Carbon and Hydrology In Dynamic EcosystEms)**

ORCHIDEE (Krinner et al., 2005) is the land component of the IPSL (Institut Pierre Simon Laplace) coupled climate model. It simulates the energy and water fluxes between the soil, the vegetation, and the atmosphere through the SECHIBA (Schématisation des Echanges Hydriques à l'Interface entre la Biosphère et l'Atmosphère, Ducoudré et al., 1993; de Rosnay and Polcher, 1998) module, while  andthe $CO_2$ fluxes and ecosystem carbon cycling throughare described by the STOMATE (Saclay Toulouse Orsay Model for the Analysis of Terrestrial Ecosystems, Viovy, 1996) module. When coupled with SECHIBA, STOMATE links the fast hydrological and biophysical processes with the carbon dynamics. STOMATE also contains a dynamic vegetation model, but this module was not activated for this study. In each grid cell, up to 12 plant functional types (PFTs) can be represented simultaneously, in addition to bare soil. LAI dynamics (from carbohydrate allocation) is simulated by STOMATE

which models the allocation of assimilates, autotrophic respiration components, foliar development, mortality and litter and soil organic matter decomposition. A factor representing drought stress (McMurtrie et al., 1990) linearly computes the rate of ribulose bisphosphate (RuBP) regeneration and the carboxylation rate.

The drought stress and the leaf age of the vegetation directly influence the photosynthetic capacity (Farquhar et al., 1980; Collatz et al., 1992; Verbeeck et al., 2011; de Weirdt et al., 2012), and  the stomatal conductance (Ball et al., 1987), which  controls the transpiration and is a function of two profiles: a fixed root density profile for each PFT, and the soil moisture profile (de Rosnay and Polcher, 1998). Canopy interception is proportional to LAI and the corresponding evaporation proceeds at potential rate, like the soil evaporation. In the latter case, however, soil moisture can become limiting if the upward diffusion to the top soil layer cannot supply enough water to sustain the required potential rate.

Soil moisture redistribution is described by a multi-layer scheme to solve the Richards equation for vertical unsaturated flow under the effect of root uptake (de Rosnay et al., 2002; Campoy et al., 2013). The hydraulic conductivity and diffusivity depend on soil moisture following the Van Genuchten (1980) model; the required parameters are taken from (Carsel and Parrish, 1988), and depend on the dominant soil texture in each grid-cell, based on the 1° × 1° texture map by Zobler (1986). The 2-m soil column is divided into 11 layers, with thickness increasing geometrically with depthwhile the saturated hydraulic conductivity exponentially decreases with depth, to account for increased compaction and reduced bioturbation (Beven and Kirkby, 1979). The precipitation rate and the soil hydraulic conductivity govern the partitioning between  surface runoff  and soil infiltration, which involves a time splitting procedure inspired from Green and Ampt (1911) to describe the propagation of the wetting front. The second contribution to total runoff is  gravitational drainage at the bottom of the soil.

The routing module (Polcher, 2003; Ngo-Duc et al., 2005; Guimberteau et al., 2012) calculates the daily discharge in each grid-cell and to the ocean. Streamflow routing relies on a series of linear reservoirs along the drainage network, derived from a 0.5° resolution data set (Vörösmarty et al., 2000). The routing scheme also includes a floodplain/swamp parameterization (d'Orgeval et al., 2008), recently improved by Guimberteau et al. (2012) for the Amazon basin, by introducing a new floodplain/swamp map. The simulation of the hydrology by the model ORCHIDEE has been widely tested over the Amazon basin and its catchments (Guimberteau et al., 2012; Getirana et al., 2014; Guimberteau et al., 2014).

| Location | Station Name | Abbreviation | River Name | Abbreviation | Latitude | Longitude | Area (km$^2$) |
|---|---|---|---|---|---|---|---|
| MAIN | Óbidos | OBI | Amazonas | AMAZ | -1.95 | -55.30 | 4,680,000 |
| SOUTH | Fazenda Vista Alegre | FVA | Madeira | MAD | -4.68 | -60.03 | 1,293,600 |
| | Guajará-Mirim | GMIR | Mamoré | MAM | -10.99 | -65.55 | 532,800 |
| | Itaituba | ITA | Tapajós | TAP | -4.24 | -56.00 | 461,100 |
| | Altamira | ALT | Xingu | XIN | -3.38 | -52.14 | 469,100 |
| WEST | Tamshiyacu | TAM | Upper Solimões | UPSO | -4.00 | -73.16 | 726,400 |
| | Lábrea | LAB | Purus | PUR | -7.25 | -64.80 | 230,000 |
| | Gavião | GAV | Juruá | JUR | -4.84 | -66.85 | 170,400 |
| NORTH | Caracaraí | CARA | Branco | BRA | +1.83 | -61.08 | 130,600 |

**Table S1.** List of the gauging stations for the studied catchments. Sources: SO HYBAM (Observation Service of the Geodynamical, hydrological and biogeochemical control of erosion/alteration and material transport in the Amazon, Orinoco and Congo basins, Cochonneau et al., 2006).

| | Basin | Model | Relative bias (%) | | Correlation coefficient | | NRMSE (%) | |
|---|---|---|---|---|---|---|---|---|
| | | | Q | ET | Q | ET | Q | ET |
| MAIN | AMAZ | INLAND-DGVM | -22.4 | -1.8 | - | 0.60 | - | 14.1 |
| | | LPJmL-DGVM | -21.9 | +1.9 | 0.77 | 0.55 | 36.6 | 25.0 |
| | | ORCHIDEE | -5.9 | -4.6 | 0.91 | 0.58 | 14.1 | 17.2 |
| SOUTH | MAD | INLAND-DGVM | -28.3 | +0.2 | - | 0.89 | - | 13.6 |
| | | LPJmL-DGVM | -2.2 | -9.5 | 0.89 | 0.83 | 33.5 | 28.9 |
| | | ORCHIDEE | -5.5 | -1.7 | 0.99 | 0.88 | 20.2 | 15.2 |
| | MAM | INLAND-DGVM | -60.9 | +0.92 | - | 0.99 | - | 15.0 |
| | | LPJmL-DGVM | +14.0 | -14.8 | 0.73 | 0.91 | 47.4 | 30.6 |
| | | ORCHIDEE | -22.2 | -3.0 | 0.91 | 0.98 | 43.4 | 18.0 |
| | TAP | INLAND-DGVM | +10.5 | -3.0 | - | 0.02 | - | 13.4 |
| | | LPJmL-DGVM | +25.1 | -6.8 | 0.90 | 0.45 | 53.9 | 34.0 |
| | | ORCHIDEE | +16.6 | -3.3 | 0.96 | 0.11 | 47.5 | 11.5 |
| | XIN | INLAND-DGVM | +47 | -1.9 | - | 0.17 | - | 14.9 |
| | | LPJmL-DGVM | +59.1 | -5.9 | 0.83 | -0.01 | 66.6 | 34.5 |
| | | ORCHIDEE | +34.1 | -4.4 | 0.94 | 0.31 | 46.1 | 12.6 |
| WEST | UPSO | INLAND-DGVM | -57.4 | +2.2 | - | 0.32 | - | 22.9 |
| | | LPJmL-DGVM | -45.2 | -0.9 | 0.93 | 0.87 | 86.0 | 18.5 |
| | | ORCHIDEE | -17.2 | -10.0 | 0.96 | 0.31 | 23.9 | 25.5 |
| | PUR | INLAND-DGVM | +9.3 | +2.6 | - | 0.83 | - | 9.8 |
| | | LPJmL-DGVM | +18.6 | +1.7 | 0.86 | 0.27 | 39.3 | 24.0 |
| | | ORCHIDEE | +15.8 | -0.9 | 0.96 | 0.79 | 31.6 | 10.1 |
| | JUR | INLAND-DGVM | +9.3 | -0.05 | - | 0.86 | - | 9.3 |
| | | LPJmL-DGVM | +10.2 | +7.3 | 0.89 | 0.02 | 29.7 | 17.0 |
| | | ORCHIDEE | +39.4 | -4.1 | 0.96 | 0.82 | 40.1 | 10.4 |
| NORTH | BRA | INLAND-DGVM | +47.1 | +17.1 | - | 0.74 | - | 21.0 |
| | | LPJmL-DGVM | +53.3 | +12.5 | 0.99 | 0.06 | 51.3 | 33.1 |
| | | ORCHIDEE | +69.3 | +10.9 | 0.96 | 0.61 | 58.8 | 15.0 |

**Table S2.** Bias (%), correlation and NRMSE (Normalized Root Mean Square Error) (%) against the observations, of discharge and ET, for each catchment, for HIST period. Observed discharge comes from SO HYBAM and ET is estimated by the product of Jung et al. (2010).

[Figure]

**Figure S1.** Deforested area (%) in each 25 x 25 km$^2$ for the LCC scenarios LODEF (**a** and **d**), HIDEF (**b** and **e**) and EXDEF (**c** and **f**).

[Figure]

**Figure S2.** Decrease of forest fraction for the three LCC scenarios (for the two time periods) compared with the NODEF scenario in 2009 over the Amazon basin. Grey colour indicates no change of forest fraction.

[Figure]

**Figure S3.** Seasonal change in ET (mm month$^{-1}$) due to deforestation combined with climate change (EXDEF) simulated by the three LSMs over the Amazon basin and its catchments, averaged over the two future periods. For a given LSM and period, the shaded area defines the envelope enclosing the range with plausible climate futures.

[Figure]

**Figure S24.** For each GCM-forcing, monthly mean seasonalities of the water budget components (including the ET components) (mm d$^{-1}$) from the three LSMs (rows) and for each NODEF and LCC scenarios (columns) over **(a)** the Madeira and **(b)** the Tapajós catchments. The variables of the water budget are: precipitation (P), runoff (R) and evapotranspiration (ET). The variables of the ET components are: transpiration (Tr), soil evaporation (Esoil) and evaporation of canopy interception (Ecanop).

---

## Editor Comment (EC1) · FF Fenicia (Editor) · 6 Dec 2016

Having read the paper and the responses of the reviewers, my consideration is that the paper has potential, but needs considerable clarifications and improvements. In particular, it is difficult to draw a line between what is the new contribution of the paper and the rest (available data, models, model results, etc.). The introduction, currently very short, should be expanded to serve this objective. The methodologies should be complemented and clarified.

Reviewer 2 mentions that the paper does not present new data or new models, and

that the contribution of the paper lies primarily in the use and integration of existing tools. Based on this, I do not understand why the paper has 19 authors. Did each of these 19 persons have a fundamental role in this research? Which one? Otherwise I would consider reducing the number of authors to a more realistic number.

---

## Author Comment (AC4) · 5 Jan 2017

Editor comments:

Having read the paper and the responses of the reviewers, my consideration is that the paper has potential, but needs considerable clarifications and improvements. In particular, it is difficult to draw a line between what is the new contribution of the paper and the rest (available data, models, model results, etc.). The introduction, currently very short, should be expanded to serve this objective. The methodologies should be complemented and clarified.

Reviewer 2 mentions that the paper does not present new data or new models, and that the contribution of the paper lies primarily in the use and integration of existing tools. Based on this, I do not understand why the paper has 19 authors. Did each of these 19 persons have a fundamental role in this research? Which one? Otherwise I would consider reducing the number of authors to a more realistic number.
* * *
Author answer:

We acknowledge your decision regarding our manuscript. We carefully addressed each of the point raised by the reviewers and we submitted an updated manuscript including many corrections and additions. Regarding your specific comments, we are willing to address them and we offer the following explanations and propositions.
* * *
C1 "considerable clarifications and improvements"

We have addressed all clarifications and improvements requested by the reviewers, namely:

- the evaluation of the models performance in present time by comparing simulated ET and discharge with dataset (new Table S2 in supplementary material)

- the discussion of the LSMs calibration which has been conducted in our answers to the reviewers 1 and 3

- a better description of the climate change scenarios (section 2.2)

- a re-written section describing the deforestation scenarios (section 2.3) and additional informations in the introduction

- a new analysis to give the different contributions of the LSM/GCM/LCC to total uncertainty in ET and runoff projections with an ANOVA (section 4.3 + new Figure 14)

We feel that the reviews have been addressed in depth and we cannot offer more clarification at this stage, unless more specific questions are raised.
* * *
C2 "draw a line between what is the new contribution of the paper"

Our manuscript is an original contribution with new LSM simulations that were not published elsewhere. The land use scenarios used as a forcing have been published by Aguiar et al. (2016) and Tejada et al. (2015) and the GCM forcing fields by Zhang et al. (2015) and Moghim et al. (2016). This is clearly detailed in the paper (sections 2.3 and 2.2 of the new version of the manuscript, respectively). In fact, reviewer 2 requested a more detailed description of the land use scenarios which has been added. To our knowledge, there is no multi-LSM study with different land use and climate scenarios for the Amazon basin rivers with a detailed attribution of uncertainties to land use change vs climate change, considering each component of the water budget of each sub basin (ET, runoff). We offer to explain this novelty of the paper more clearly in the revised introduction.
* * *
C3 "The introduction, currently very short, should be expanded to serve this objective".

We revised and expanded the introduction as explained above.
* * *
C4 "The methodologies should be complemented and clarified".

We already addressed this in the revised manuscript with a complement about the land use scenarios in section 2.3, the evaluation of models in Table S2 and their descriptions are better described in the Supplementary material.
* * *
C5 "Reviewer 2 mentions that the paper does not present new data or new models, and that the contribution of the paper lies primarily in the use and integration of existing tools. Based on this, I do not understand why the paper has 19 authors."

Reviewer 2 is right that the paper is based on published land use scenarios and GCM forcing but there was a significant work to adapt these scenarios and forcings to the objectives of the AMAZALERT project and to the LSMs. The novelty is in performing the simulations with a new protocol and separating the effects of land use vs climate change in different sub-basins. Integration studies are very much needed for climate assessments in regions like the Amazon where both land use change and climate change are susceptible to modify hydrological variables, and our study is the first to date to offer a formal attribution of uncertainties to land use vs climate change vs LSM structure. As a result, because 3 LSMs with LCC and CC scenarios are involved through the AMAZALERT project, we have the following co-authors, who all contributed to write the manuscript:

ORCHIDEE model

- Matthieu Guimberteau (in charge of the whole hydrological study and performed the model simulations)

- Philippe Ciais (coordinator for the IPSL work (ORCHIDEE modeling) in the AMAZALERT project)

- Agnès Ducharne (main contributor of the hydrology modeling)

- Juan Pablo Boisier (responsible of the LCC modeling)

- Hans Verbeeck (responsible of the AMAZALERT modeling group and contributor of the vegetation modeling in ORCHIDEE)

————————————

LPJmL-DGVM model

- Hester Biemans (contributor of the hydrology modeling)

- Fanny Langerwisch (contributor of the hydrology modeling and performed the model simulations)

- Anja Rammig and Kirsten Thonicke (both contributors of the hydrology-vegetation modeling)

———————————

INLAND-DGVM model

- Daniel Andres Rodriguez and Rita C.S. von Randow (both contributors of the hydrology modeling)

- Celso von Randow (contributor of the hydrology modeling and performed the model simulations)

———————————

LCC scenarios

- Ana Paula Dutra Aguiar (built the Brazilian LCC scenarios)

- Graciela Tejada (built the Bolivian LCC scenarios)

———————————

LSMs quality control

- Hannes De Deurwaerder (post-processing and quality control of the three LSMs)

———————————

Climate change scenarios

- Ke Zhang (built the climate change scenarios for the Amazon applying downscaling methodology)

- David Galbraith (contributor of the protocol adopted for this study)

————————————

Articulation of the present study with the rest of the AMAZELRT project

- Bart Kruijt (coordinator of the AMAZALERT project)

- German Poveda (in charge of studying the effect of climate on the hydrology)

---

## Author Response (AR2)

**Comments of one of the reviewer:**

*Dear authors,*

*I am satisfied with the changes by the authors and think the paper is much improved. In my opinion the new section which analyzes the different contributions of the various LSMs/GCMs/LCCs to total uncertainty in ET and runoff projections (section 4.3 + new Figure 14) is the most interesting and transferable part of the study. I recommend that the abstract be revised to better summarize the big picture findings of this work rather than focus on the numerical results, i.e. include more from the synthesis and discussion section, and less from the pure results section. In my opinion the findings of the differences in relative uncertainties of LSM/GCM/LCC by process and by region are very interesting and deserve to be included in the abstract. Thus my recommendation is to accept the paper, subject to minor revisions to the abstract.*

**Answers of the authors**

We re-wrote the abstract which includes now less numerical results and more elements of the discussion, in particular of the contributions of the various factors to total uncertainties in ET and runoff projections.
We also re-organized the section 4.3 for better comprehension of the results obtained from the uncertainty analyse.